# DRIVINGRECON: LARGE 4D GAUSSIAN RECONSTRUCTION MODEL FOR AUTONOMOUS DRIVING

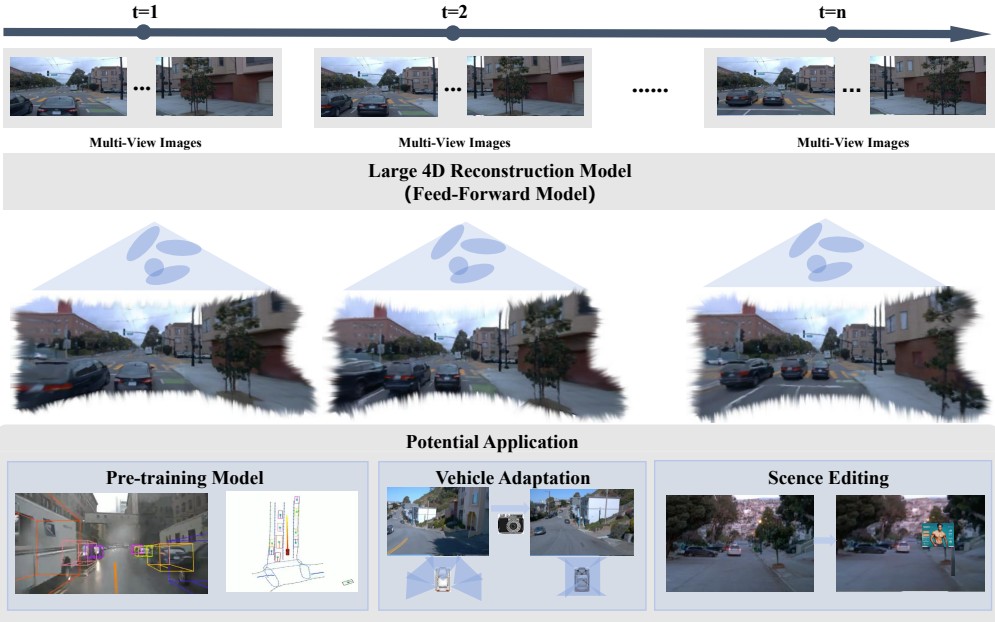

Figure 1: The overview. Leveraging temporal multi-view images, the Large 4D Gaussian Reconstruction Model (DrivingRecon) is capable of predicting 4D driving scenes. DrivingRecon serves as a pre-trained model that effectively captures geometric and motion information, thereby enhancing performance in perception, tracking, and planning tasks. Additionally, DrivingRecon can synthesize novel views based on specific camera parameters, ensuring adaptability to various vehicle models. Furthermore, DrivingRecon facilitates the editing of designated 4D scenes through the removal, insertion, and manipulation of objects.

## ABSTRACT

Photorealistic 4D reconstruction of street scenes is essential for developing real-world simulators in autonomous driving. However, most existing methods perform this task offline and rely on time-consuming iterative processes, limiting their practical applications. To this end, we introduce the Large 4D Gaussian Reconstruction Model (DrivingRecon), a generalizable driving scene reconstruction model, which directly predicts 4D Gaussian from surround-view videos. To better integrate the surround-view images, the Prune and Dilate Block (PD-Block) is proposed to eliminate overlapping Gaussian points between adjacent views and remove redundant background points. To enhance cross-temporal information, dynamic and static decoupling is tailored to learn geometry and motion features better. Experimental results demonstrate that DrivingRecon significantly improves scene reconstruction quality and novel view synthesis compared to existing methods. Furthermore, we explore applications of DrivingRecon in model pre-training, vehicle adaptation, and scene editing. Our code will be made publicly available.

# 1 INTRODUCTION

Autonomous driving has made remarkable advancements in recent years, particularly in the areas of perception (Li et al., 2022b; Zhang et al., 2022; Huang et al., 2023; Wei et al., 2023b), prediction (Hu et al., 2021; Gu et al., 2022; Liang et al., 2020), and planning (Dauner et al., 2023; Cheng et al., 2022; 2023; Hu et al., 2023). With the emergence of end-to-end autonomous driving systems that directly derive control signals from sensor data (Hu et al., 2022; 2023; Jiang et al., 2023), conventional open-loop evaluations have become less effective (Zhai et al., 2023; Li et al., 2024). Real-world closed-loop evaluations offer a promising solution, where the key lies in the development of high-quality scene reconstruction (Turki et al., 2023; Xie et al., 2023).

Despite numerous advancements in the photo-realistic reconstruction of small-scale scenes (Mildenhall et al., 2021; Müller et al., 2022; Chen et al., 2022; Kerbl et al., 2023; Wei et al., 2023a), modeling large-scale and dynamic driving environments remains challenging. Most existing methods tackle these challenges by using 3D bounding boxes to differentiate static from dynamic components (Yan et al., 2024; Wu et al., 2023b; Turki et al., 2023). Subsequent methods learn the dynamics in a self-supervised manner with a 4D NeRF field (Yang et al., 2023a) or 3D displacement field (Huang et al., 2024). The aforementioned methods require numerous and time-consuming iterations for reconstruction and cannot generalize to new scenes.

While some recent methods are able to reconstruct 3D objects (Hong et al., 2023; Zhang et al., 2024; Tang et al., 2024) or 3D indoor scenes (Charatan et al., 2024; Chen et al., 2024; Szymanowicz et al., 2024) with a single forward pass, these approaches are not directly applicable to dynamic driving scenarios. Specifically, two core challenges arise in driving scenarios: (1) Models tend to predict redundant Gaussian points across adjacent views, leading to model collapse. (2) At a given moment, the scene is rendered with a very limited supervised view (sparse view supervision), and the presence of numerous dynamic objects limits the direct use of images across time sequences.

To this end, we introduce a Large Spatial-Temporal Gaussian Reconstruction Model (DrivingRecon) for autonomous driving. Our method starts with a 2D encoder that extracts image features from surround-view images. A DepthNet module estimates depth to derive world coordinates using camera parameters. These coordinates, along with the image features, are fed into a temporal cross-attention mechanism. Subsequently, a decoder integrates this information with additional Prune and Dilate Blocks (PD-Blocks) to enhance multi-view integration. The PD-Block effectively prunes overlapping Gaussian points between adjacent views and redundant background points. The pruned Gaussian points can be replaced by dilated Gaussian points of complex object. Finally, a Gaussian Adapter predicts Gaussian attributes, offsets, segmentation, and optical flow, enabling dynamic and static object rendering. By leveraging cross-temporal supervision, we effectively address the sparse view challenges. Our main contributions are as follows:

- To the best of our knowledge, we are the first to explore a feed-forward 4D reconstruction model specifically designed for surround-view driving scenes.

- We propose the PD-Block, which learns to prune redundant Gaussian points from different views and background regions. It also learns to dilate Gaussian points for complex objects, enhancing the quality of reconstruction.

- We design rendering strategies for both static and dynamic components, allowing rendered images to be efficiently supervised across temporal sequences.

- We validate the performance of our algorithm in reconstruction, novel view synthesis, and cross-scene generalization.

- We explore the effectiveness of DrivingRecon in pre-training, vehicle adaptation, and scene editing tasks.

# 2 RELATED WORK

## 2.1 DRIVING SCENE RECONSTRUCTION

Numerous efforts have been put into reconstructing scenes from autonomous driving data captured in real scenes. Existing self-driving simulation engines such as CARLA (Dosovitskiy et al., 2017)

or AirSim (Shah et al., 2017) suffer from costly manual effort to create virtual environments and the lack of realism in the generated data. Many studies have investigated the application of these methods for reconstructing street scenes. Block-NeRF (Tancik et al., 2022) and Mega-NeRF (Turki et al., 2021) propose segmenting scenes into distinct blocks for individual modeling. Urban Radiance Field (Rematas et al., 2021) enhances NeRF training with geometric information from LiDAR, while DNMP (Lu et al., 2023) utilizes a pre-trained deformable mesh primitive to represent the scene. Streetsurf (Guo et al., 2023) divides scenes into close-range, distant-view, and sky categories, yielding superior reconstruction results for urban street surfaces. MARS (Wu et al., 2023b) employs separate networks for modeling background and vehicles, establishing an instance-aware simulation framework. With the introduction of 3DGS (Kerbl et al., 2023b), DrivingGaussian (Zhou et al., 2023) introduces Composite Dynamic Gaussian Graphs and incremental static Gaussians, while StreetGaussian (Yan et al., 2024) optimizes the tracked pose of dynamic Gaussians and introduces 4D spherical harmonics for varying vehicle appearances across frames. Omnire (Chen et al., 2024) further focus on the modeling of non-rigid objects in driving scenarios. However, these reconstruction algorithms requires time-consuming iterations to build a new scene.

## 2.2 LARGE RECONSTRUCTION MODELS

Some works have proposed to greatly speed this up by training neural networks to directly learn the full reconstruction task in a way that generalizes to novel scenes Yu et al. (2021); Wang et al. (2021; 2022); Wu et al. (2023a). Recently, LRM (Hong et al., 2023) was among the first to utilize large-scale multiview datasets including Objaverse (Deitke et al., 2023) to train a transformer-based model for NeRF reconstruction. The resulting model exhibits better generalization and higher quality reconstruction of object-centric 3D shapes from sparse posed images in a single model forward pass. Similar works have investigated changing the representation to Gaussian splatting (Tang et al., 2024; Zhang et al., 2024), introducing architectural changes to support higher resolution (Xu et al., 2024; Shen et al., 2024), and extending the approach to 3D scenes (Charatan et al., 2023; Chen et al., 2024). Recently, L4GM utilize temporal cross attention to fuses multiple frame information to predict the Gaussian representation of a dynamic object (Ren et al., 2024). However, for autonomous driving, there is no one to explore the special method to fuse surround-views. The naive model predicts repeated Gaussian points of adjacent views, significantly reducing reconstruction performance. Besides, sparse view supervision and numerous dynamic objects further complicate the task.

## 3 METHOD

In this section, we present the Large 4D Reconstruction Model (DrivingRecon), which generates 4D scenes from surround-view video inputs in a single feed-forward pass. Section 3.1 details the overview of DrivingRecon. In Section 3.2, we provide an in-depth examination of the Prune and Dilate Block (PD-Block). Finally, Section 3.3 discusses our training strategy, which includes static and dynamic decoupling, 3D-aware positional encoding, and segmentation techniques.

### 3.1 OVERALL FRAMEWORK

**Symbol definition.** DrivingRecon utilizes temporal multi-view images $D$ to train a feedforward model $\mathbf{G} = f(D)$. This model predicts Gaussians $\mathcal{G} = \{\mathbf{G} \in \mathbb{R}^d\}$ in the structure of $(\mathbf{xyz} \in \mathbb{R}^3, \mathbf{rgb} \in \mathbb{R}^3, \mathbf{a} \in \mathbb{R}^1, \mathbf{s} \in \mathbb{R}^3, \mathbf{c} \in \mathbb{R}^{|\mathcal{C}|}, \mathbf{r} \in \mathbb{R}^4, \mathbf{\Delta xyz} \in \mathbb{R}^3, \mathbf{\Delta r} \in \mathbb{R}^4)$. These elements represent position, RGB color, scale, rotation vectors, semantic logits, position change and rotation change, respectively. For the $i$-th sample, $D^i = \{X^t, R^t, V^t, E^t \mid t = 1, \ldots, T\}$ includes $N$ multi-view images $X^t = \{I_1, \ldots, I_j, \ldots, I_N\}$ at each timestep $t$, with corresponding intrinsic parameters $\mathcal{E}^t = \{E_1, \ldots, E_j, \ldots, E_N\}$, extrinsic rotation $\mathcal{R}^t = \{R_1, \ldots, R_j, \ldots, R_N\}$, and extrinsic translation $\mathcal{V}^t = \{V_1, \ldots, V_j, \ldots, V_N\}$. The extrinsic parameter is to project the camera coordinate system directly into the world coordinate system. We take the video start frame as the origin of the world coordinate system.

**Pipeline.** The temporal multi-view images $D$ are processed through a shared image encoder $F_{img}$ to extract image features $e_{img}$. A specialized 3D Position Encoding method leverages a Depth-Net alongside camera intrinsic and extrinsic parameters to compute the world coordinates $(x, y, z)$. These coordinates are concatenated with the image features $e_{img}$ to form geometry-aware features

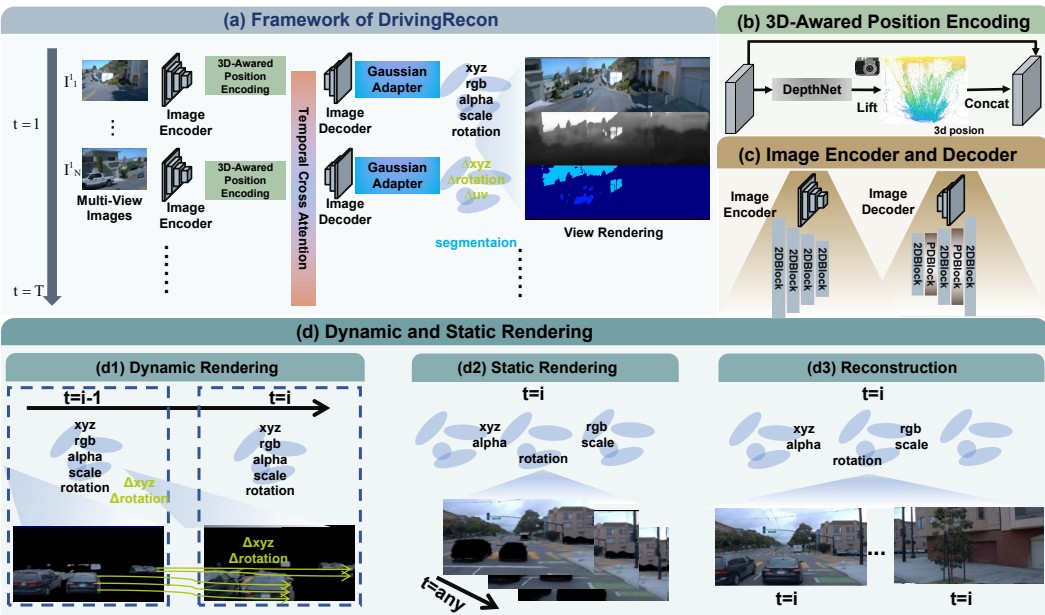

Figure 2: The overview of DrivingRecon. (a) Multi-view images are in turn sent to encoder, 3D-aware positional encoding, temporal cross-attention, decoder, and Gaussian adaptor to directly predict 4D Gaussians. (b) The 3D-aware Positional Encoding (3D-PE) leverages DepthNet, alongside camera parameters, to compute 3D world coordinates. These coordinates are integrated with the image features to enhance geometry awareness. (c) The visual encoder comprises multiple 2D convolutional blocks, while the visual decoder includes both 2D convolutional blocks and PD-Blocks. Details of the PD-Block are provided in Sec. 3.2. (d) For dynamic objects, we only use next time-step images to supervise the current Gaussian parameters. For static scenes, rendering supervision is used across timestamps. In addition, reconstruction loss is also applied.

$e_{geo}$. Then, temporal cross-attention merge features from different timesteps. The decoder then enhances the resolution of these image features. Finally, a Gaussian adapter transforms the decoded features into Gaussian points and segmentation outputs. In the decoder, the Prune and Dilate block (PD-Block) can integrate image features from various viewpoints. It is worth mentioning that we used the UNet structure, which is not shown in the Figure

**3D Position Encoding.** To better integrate features across different views and time intervals, we implement 3D position encoding. Our DepthNet predicts feature depth $d_{u,v}$ at UV-coordinate positions $(u, v)$. This involves a straightforward operation: selecting the first channel of the image feature and applying the Tanh activation function to predict depths. The predicted depths $d_{u,v}$ is subsequently converted into world coordinates $[x, y, z] = R \times E^{-1} \times d_{u,v} \times [u, v, 1] + V$. These coordinates are directly concatenated with the image features for input into the PD-Block, enabling multi-view feature fusion.

**Temporal Cross Attention.** Due to the sparse nature of multi-view data with minimal overlap, neural networks face challenges in comprehending the geometric information of scenes and objects. By fusing multiple timestamps, we effectively integrate more viewing angles, enhancing the modeling of scene geometry and understanding of both static and dynamic objects. Temporal self-attention is employed to merge temporal features by considering both temporal and spatial dimensions simultaneously, as detailed in (Ren et al., 2024).

**Gaussian Adapter.** The Gaussian adapter employs two convolutional blocks to convert features into segmentation $\mathbf{c} \in \mathbb{R}^C$, depth categories $\mathbf{d_c} \in \mathbb{R}^L$, depth regression refinement $\mathbf{d_r} \in \mathbb{R}^1$, RGB color $\mathbf{rgb} \in \mathbb{R}^3$, alpha $\mathbf{a} \in \mathbb{R}^1$, scale $\mathbf{r} \in \mathbb{R}^3$, rotation $\mathbf{r} \in \mathbb{R}^3$, UV-coordinate shifts $[\Delta u, \Delta v]$, and optical flow $[\Delta x, \Delta y, \Delta z]$. The activation functions for RGB color, alpha, scale, and rotation are consistent with those in (Tang et al., 2024). The final depth per pixel is computed as $\mathbf{d_f} = \sum_{l=1}^{L} l \times \text{softmax}(\mathbf{d_c}) + \mathbf{d_r}$. The UV-coordinate shifts $[\Delta u, \Delta v]$ indicate that our approach is not strictly pixel-aligned for Gaussian prediction, as elaborated in Sec. 3.2.

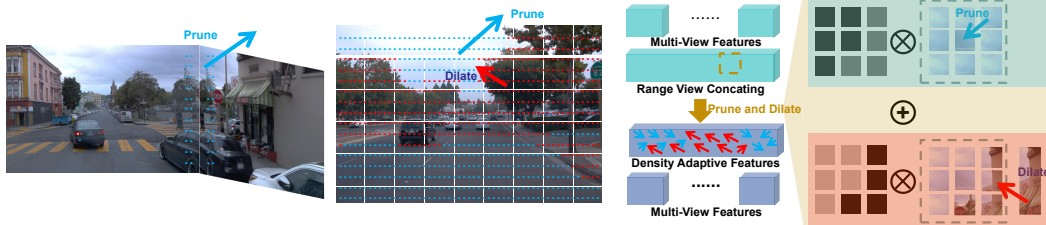

(a) Repetition of Different Views     (b) Redundant Background     (c) Prune and Dilate Block

Figure 3: The motivation and details of Prune and Dilate Block (PD-Block). (a) Different views predict repeated Gaussian points, causing the model collapse. (b) Simple backgrounds (blue dots) do not need a large number of Gaussian dots to be represented, while complex objects (red dots) need more Gaussian dots to be represented. (c) PD-Block fuse the multi-view image features into a range view form. Then PD-Block prune and dilate the Gaussian points according to the complexity of the scene.

## 3.2 Learn to Prune and Dilate

There are two core problems with surround-view driving scene reconstruction: (1) Overlapping parts of different view angles will predict repeated Gaussian points, and these repeated Gaussian points will cause the collapsion as shown in Fig 3 (a). (2) The edge of an object that is too complex often requires more Gaussian points to describe it, while the sky and the road are very similar and do not need too many Gaussian points as shown in Fig 3 (b).

**Prune and Dilate Block.** To this end, we propose a Prune and Dilate Block (PD-Block), which can dilate the guassian point of complex instances and prune the gaussian of similar backgrounds or different-views as shown in Figure 2 (c). (1) First, we directly concatenate the adjacent image features in the form of a range view (Kong et al., 2023), in other words, to make the overlapping parts of the 3D position easier to merge. (2) Then we cut the range view feature into multiple region, which can greatly reduce the memory usage. (3) Following (Achanta et al., 2012; Ma et al., 2023), we evenly propose $K$ centers in space, and the center feature is computed by averaging its $Z$ nearest points. (4) We then calculate the pair-wise cosine similarity matrix $S$ between the region feature and the center points. (5) We set a threshold $\tau$ to generate a mask $M$ that is considered 0 if it is below this threshold and 1 if it is above this threshold. In addition, the point most similar to the center has always been retained. (6) Based on mask, we can aggregate the long-term features $e_{lt}$ and the local features $e_{lc}$, $e = M * e_{lt} + (1 - M) * e_{lc}$. Here, the long-term features $e_{lt}$ is extracted by a large kernal convolution, and the local features $e_{lc}$ is the original range view features.

**Unaligned Gaussian Points.** PD-Blocks effectively manage spatial computational redundancy by reallocating resources from simple scenes to more complex objects, allowing for Gaussian points that are not strictly pixel-aligned.For this reason, our Guassian Adapter also predicts the offset of the uv coordinate $[\Delta u, \Delta v]$ as described in Sec. 3.1. The world coordinate $[x, y, z] = RE^{-1}\mathbf{d_f} * [u + \Delta u, v + \Delta v, 1] + V$. The above operations are universal for any time and view, so we did not label the time and views for simplicity. Gaussian points of different viewing angles are all fused to render. In addition, we can use the world coordinates at time t and the predicted optical flow to get the world coordinates at time t+1, $[x_{t+1}, y_{t+1}, z_{t+1}] = [x_t + \Delta x_t, y_t + \Delta y_t, z_t + \Delta z_t]$. Rotational changes in an object are interpreted as positional changes.

## 3.3 Training Objective

To learn geometry and motion information, DrivingRecon carefully designed a series of regulations, including segmentation regulation, dynamic and static rendering regulation, and 3D-aware coding regulation.

**Static and Dynamic Decoupling.** The views of the driving scene are very sparse, meaning that only a limited number of cameras capture the same scene simultaneously. Hence, cross-temporal view supervision is essential. For dynamic objects, our algorithm predicts not only the current Gaussian of dynamic objects at time $t$ but also predicts the flow of each Gaussian point. Therefore, we will also use the next frame to supervise the predicted Gaussian points, i.e., $\mathcal{L}_{dr}$. For static objects, we can render the scene with camera parameters of adjacent timestamps and supervise only the static part, i.e., $\mathcal{L}_{sr}$. Most algorithms only use static object scenes to better build 3D Gauss, neglecting the

supervision of multiple views of dynamic objects. It is important to note that when supervising the rendering across the time sequence, we will not supervise the rendered image where the threshold value is less than $\alpha$, as these pixels often do not overlap across the time sequence. Additionally, we have the L1 reconstruction constraint $\mathcal{L}_{re}$, which involves rendering the image as the same as the input.

**3D-aware Position Encoding Regulation.** Accurate 3D position encoding allows for better fusion of multiple views (Shu et al., 2023). In Section 3.1, we introduced 3D position encoding. Here, we explicitly supervise the depth $d_{u,v}$ with regulation loss $\mathcal{L}_{PE} = M_d |d_{u,v}^{gt} - d_{u,v}|$. Here, $d_{u,v}^{gt}$ represents the depth of the 3D point cloud projected onto the UV plane, and $M_d$ is the mask indicating the presence of a LiDAR point.

**Segmentation.** Segmentation supervision can help the network better understand the semantics of the scene and can also decompose static objects for cross-temporal view supervision. We utilize the DeepLabv3plus to produce three kinds of masks: dynamic objects (various vehicles and people), static objects, and the sky [1]. Additionally, we project a 3D box onto a 2D plane as a prompt to use SAM to generate more accurate dynamic object masks. The masks of two dynamic objects are fused using "or" logic to ensure that all dynamic objects are masked. Cross-entropy loss is used to constrain the segmentation results predicted by Gaussian Adapter, i.e., $\mathcal{L}_{seg}$. We also employ cross-entropy loss $\mathcal{L}_c$ for the depth categories $_c$ predicted by the Gaussian Adapter and L1 loss $\mathcal{L}_r$ for the refined depth $_r$. In summary, the overall constraints for training DrivingRecon are:

$$\mathcal{L}_{total} = \lambda_{re}\mathcal{L}_{re} + \lambda_c\mathcal{L}_c + \lambda_r\mathcal{L}_r + \lambda_{PE}\mathcal{L}_{PE} + \lambda_{dr}\mathcal{L}_{dr} + \lambda_{sr}\mathcal{L}_{sr} + \lambda_{seg}\mathcal{L}_{seg}$$

where each $\lambda$ term balances the contribution of the respective loss component. $\mathcal{L}_{sr}$ and $\mathcal{L}_{seg}$ used segmentation labels, which is not used for pre-training experiment. Other loss are considered unsupervised, which also allows DrivingRecon to achieve good performance. These collective regulations and constraints enable DrivingRecon to effectively integrate geometry and motion information, enhancing its capacity for accurate scene reconstruction across time and perspectives.

## 4 EXPERIMENT

In this section, we evaluate the performance of DrivingRecon in terms of reconstruction and novel view synthesis, as well as explore its potential applications. We also provide detailed information on the dataset setup, baseline methods, and implementation details.

**Datasets.** The NOTR dataset is a subset of the Waymo Open dataset (Sun et al., 2020) curated by (Yang et al., 2023a).The Diverse-56 dataset comprises various challenging driving scenarios, including ego-static, dusk/dawn, gloomy, exposure mismatch, nighttime, rainy, and high-speed, which will be used to evaluate the algorithm's performance across different scenarios. To create a balanced and diverse standard dataset, we combine the NOTR's dynamic32 (D32) and static32 (S32) datasets to form NOTA-DS64. Additionally, the nuScenes (Caesar et al., 2020) dataset is utilized to test the algorithm's adaptability to downstream tasks.

**Training Details.** The model is trained on 24 NVIDIA A100 (80G) GPUs for 50000 iterations. A batch size of 2 for each GPU is used under bfloat16 precision, resulting in an effective batch size of 48. The input resolution of DrivingRecon is $256 \times 512$. We trained the model using multiple views of three consecutive moments. The AdamW optimizer is employed with a learning rate of $4 * 10^{-4}$ and a weight decay of 0.05. $\lambda_{re}, \lambda_c, \lambda_r, \lambda_{PE}, \lambda_{dr}, \lambda_{sr}, \lambda_{seg}$ are set as 1.0, 0.1, 0.1, 0.1, 0.1, 0.1, 0.1, respectively. These balance parameters are based on our experience.

### 4.1 IN-SCENE EVALUATION

We conduct in-scene evaluations on Waymo-DS64. We select the state-of-the-art methods LGM (Tang et al., 2024), pixelSplat (Charatan et al., 2023), MVSPlat (Chen et al., 2024), and L4GM (Ren et al., 2024) as Baseline. All the algorithms incorporate depth supervision. Following the approach of (Yang et al., 2023a; Huang et al., 2024), we assess the quality of both reconstruction and novel view synthesis. We sample at intervals of 10 as labels for novel view synthesis, and these

---

[1] https://github.com/VainF/DeepLabV3Plus-Pytorch

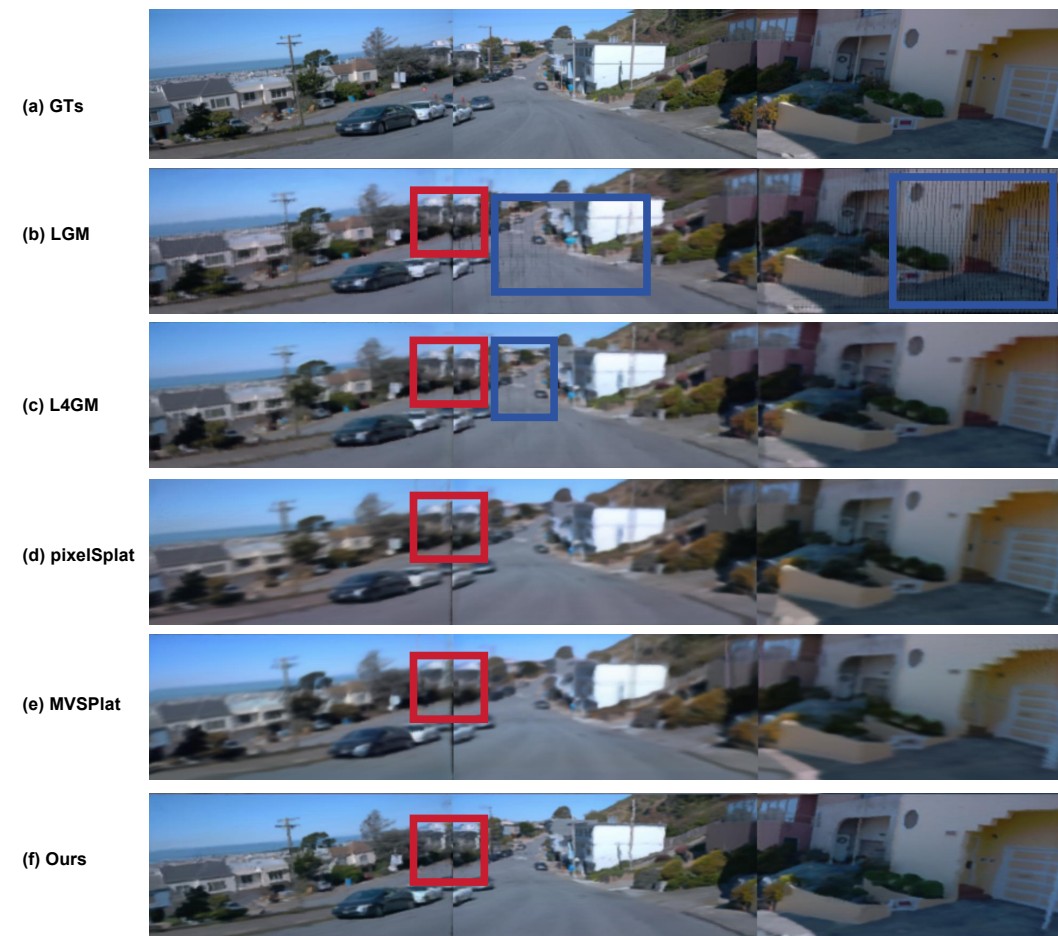

Figure 4: The qualitative comparison of reconstruction performance. The blue box indicates that there will be a large number of empty areas without Gaussian points. The red areas indicate areas where our approach is clear across perspectives.

| Method | PSNR | SSIM | LPIPS | PSNR (Static) | SSIM (Static) | PSNR (Dynamic) | SSIM (Dynamic) |
|---|---|---|---|---|---|---|---|
| LGM | 19.52 | 0.52 | 0.32 | 19.60 | 0.50 | 17.71 | 0.41 |
| pixelSplat | 20.54 | 0.58 | 0.28 | 20.76 | 0.57 | 18.11 | 0.49 |
| MVSplat | 21.33 | 0.64 | 0.24 | 21.64 | 0.61 | 19.80 | 0.53 |
| L4GM | 20.01 | 0.54 | 0.30 | 20.69 | 0.54 | 17.35 | 0.44 |
| **Ours** | **23.70** | **0.68** | **0.17** | **24.09** | **0.69** | **21.50** | **0.56** |

Table 1: Reconstruction performance on Waymo NOTA-DS64.

samples are not used for training. During testing, we do not have access to these data, but use the Gaussian predicted by the adjacent image to render these novel views.

As indicated in Table 1 and Table 2, our algorithm demonstrates significant improvements in both reconstruction and novel view synthesis. Moreover, there is a notable enhancement in the reconstruction of both static and dynamic objects, particularly dynamic objects, as we leverage timing information to predict the movement of objects.

Furthermore, we provide a visualization of the reconstruction to further illustrate the validity of our approach. As depicted in Fig 4, there are some missing areas in the reconstructions from LGM and L4GM, attributed to the challenge of directly predicting xyz relative to predicting depth. In areas with overlapping views, our algorithm displays a substantial improvement compared to any other algorithm, indicating that our PD-Block effectively integrates information from multiple view angles and eliminates redundant Gaussian points. Additionally, we visualize the ability of our method to render new views, as shown in Figure 5.

**Different rendering views**

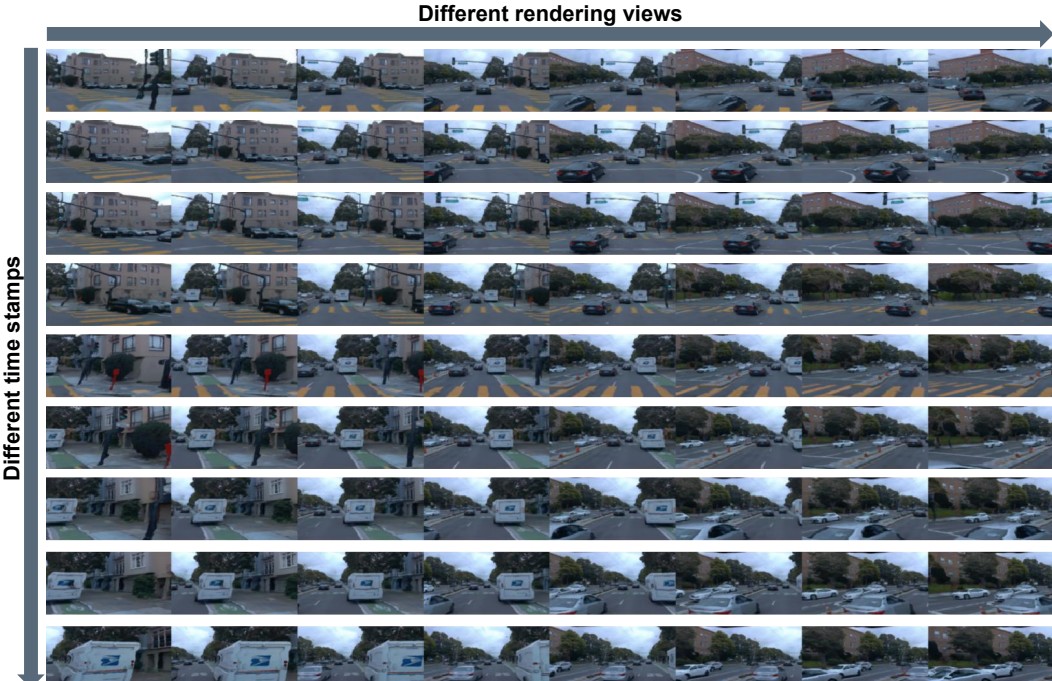

**Different time stamps**

Figure 5: Novel view rendering. Based on the predicted Gaussians, we render different views at different times. The novel views are of very high quality and very high spatio-temporal consistency **(zoom in for the best view.)**

| Method | PSNR | SSIM | LPIPS | PSNR (Static) | SSIM (Static) | PSNR (Dynamic) | SSIM (Dynamic) |
|---|---|---|---|---|---|---|---|
| LGM | 17.49 | 0.47 | 0.33 | 17.79 | 0.49 | 15.37 | 0.39 |
| pixelSplat | 18.24 | 0.56 | 0.30 | 18.63 | 0.58 | 16.96 | 0.44 |
| MVSplat | 19.00 | 0.57 | 0.28 | 19.29 | 0.58 | 17.35 | 0.47 |
| L4GM | 17.63 | 0.54 | 0.31 | 18.58 | 0.56 | 16.78 | 0.43 |
| **Ours** | **20.63** | **0.61** | **0.21** | **20.97** | **0.62** | **19.70** | **0.51** |

Table 2: Novel view synthesis evaluation on Waymo NOTA-DS64.

## 4.2 CROSS-SCENE EVALUATION

Our algorithm demonstrates strong generalization performance, as it can directly model new scenes in 4D. To validate the effectiveness of our algorithm, we utilized the model trained on NOTA-DS64 to perform reconstruction and novel view evaluation on Diverse-56 , as presented in Tab 3. The results indicate that our algorithm performs well in more challenging and even unseen scenarios. Specifically, compared with Tab 1 and Table 2, the performance of reconstruction and novel view synthesis is not significantly reduced, further emphasizing the generalization capability of our algorithm.

## 4.3 ABLATION STUDY

To assess the effectiveness of our proposed algorithm, we conducted a series of ablation experiments. The key components under evaluation include the PD-Block, Dynamic and Static Rendering (DS-R), 3D-Aware Position Encoding (3D-PE), and Temporal Cross Attention (TCA). Each of these components plays a critical role in the overall performance of the model.

As shown in Table 4a, each module contributes significant performance improvements. Notably, the PD-Block achieves the highest enhancement. This improvement stems from two primary factors: (1) an optimized distribution of computational resources based on spatial complexity, where more Gaussian points are allocated to complex regions while simpler backgrounds receive fewer points; (2) enhanced multi-perspective integration within a broad field of view. The DS-R mechanism also led to marked improvements, largely attributed to the use of cross-temporal supervision for better dynamic and static object differentiation. The 3D-Aware Position Encoding (3D-PE) facilitates the

| | Reconstruction | | | Novel View | | |
|---|---|---|---|---|---|---|
| Method | PSNR | SSIM | LPIPS | PSNR | SSIM | LPIPS |
| LGM | 16.80 | 0.44 | 0.39 | 17.94 | 0.43 | 0.42 |
| pixelSplat | 19.26 | 0.51 | 0.35 | 18.53 | 0.48 | 0.39 |
| MVSplat | 20.53 | 0.54 | 0.34 | 19.63 | 0.52 | 0.36 |
| L4GM | 19.69 | 0.51 | 0.35 | 18.92 | 0.49 | 0.38 |
| **Ours** | **22.73** | **0.65** | **0.21** | **21.41** | **0.57** | **0.26** |

Table 3: The performance of reconstruction and novel view synthesis generalization ability in new scenes (tested on Diversity-54).

| | PSNR | SSIM | LPIPS |
|---|---|---|---|
| all | 22.73 | 0.65 | 0.21 |
| w/o PD-Block | 19.27 | 0.50 | 0.36 |
| w/o DS-R | 21.44 | 0.59 | 0.27 |
| w/o 3D-PE | 21.65 | 0.60 | 0.25 |
| w/o TCA | 20.10 | 0.55 | 0.31 |

| Training Num. | PSNR | SSIM | LPIPS |
|---|---|---|---|
| 32 | 21.47 | 0.54 | 0.31 |
| 64 | 22.85 | 0.63 | 0.21 |
| 128 | 23.97 | 0.67 | 0.18 |
| 256 | 24.10 | 0.69 | 0.17 |
| 512 | 24.25 | 0.71 | 0.16 |

(a) Ablation of DrivingRecon.      (b) Scaling up ability of DrivingRecon.

Table 4: Ablation study and scaling up experiments (tested on Diversity-54).

| Waymo → nuScenes | Target Domain (nuScenes) | | | | |
|---|---|---|---|---|---|
| Method | mAP↑ | mATE↓ | mASE↓ | mAOE↓ | NDS* ↑ |
| Oracle | 0.475 | 0.577 | 0.177 | 0.147 | 0.587 |
| DG-BEV | 0.303 | 0.689 | 0.218 | 0.171 | 0.472 |
| PD-BEV | 0.311 | 0.686 | 0.216 | 0.170 | 0.478 |
| Ours* | 0.305 | 0.690 | 0.219 | 0.167 | 0.471 |
| **Ours*+** | **0.323** | **0.675** | **0.212** | **0.166** | **0.490** |

Table 5: Comparison of different approaches on domain generalization protocols, where * stands for using aligned intrinsic parameters, + stands for randomly augmenting camera extrinsic parameters.

network's ability to learn geometric information in advance, thereby improving the effectiveness of subsequent multi-view fusion. Temporal Cross Attention (TCA) further strengthens the model by efficiently incorporating temporal information, which enhances the learning of both geometric and motion-related features.

In addition to these core components, scalability was also a focus of our study. We investigated the impact of varying the number of training samples, excluding the Diverse-56 dataset, by randomly sampling from Waymo's dataset with sizes ranging from 32 to 512 samples. As shown in the Table 4b, performance gradually improves as the number of training samples increases. It is noteworthy that even with a small number of samples, our algorithm shows strong generalization.

### 4.4 POTENTIAL APPLICATION

**Vehicle adaptation.** The introduction of a new car model may result in changes in camera parameters, such as camera type (intrinsic parameters) and camera placement (extrinsic parameters). The 4D reconstruction model is capable of rendering images with different camera parameters to mitigate the potential overfitting of these parameters. To achieve this, we rendered images on Waymo with random intrinsic parameters and performed random rendering of novel views as a form of data augmentation. It is important to note that our rendered images also undergo an augmentation pipeline as part of the detection algorithm, including resizing and cropping. Subsequently, we used this jointly rendered and original data to train the BEVDepth on Waymo, following the approach of (Wang et al., 2023; Lu et al., 2023).

As demonstrated in Table 5, when we employ both camera intrinsic and extrinsic parameter augmentation, we observe a significant improvement in performance. However, the use of only camera intrinsic parameter augmentation did not yield good results, due to the superior ability of virtual

| Method | Detection | | Tracking | | | Future Occupancy Prediction | | | |
|---|---|---|---|---|---|---|---|---|---|
| | NDS ↑ | mAP ↑ | AMOTA↑ | AMOTP↓ | IDS↓ | IoU-n.↑ | IoU-f.↑ | VPQ-n.↑ | VPQ-f.↑ |
| UniAD | 49.36 | 37.96 | 38.3 | 1.32 | 1054 | 62.8 | 40.1 | 54.6 | 33.9 |
| ViDAR | 52.57 | 42.33 | 42.0 | 1.25 | 991 | 65.4 | 42.1 | 57.3 | 36.4 |
| **Ours+** | **53.21** | **43.21** | **42.9** | **1.18** | **948** | **66.5** | **43.3** | **58.2** | **37.3** |

| Method | Mapping | | Motion Forecasting | | | Planning | |
|---|---|---|---|---|---|---|---|
| | IoU-lane↑ | IoU-road↑ | minADE↓ | minFDE↓ | MR↓ | avg.L2↓ | avg.Col.↓ |
| UniAD | 31.3 | 69.1 | 0.75 | 1.08 | 0.158 | 1.12 | 0.27 |
| ViDAR | 33.2 | 71.4 | 0.67 | 0.99 | 0.149 | 0.91 | 0.23 |
| **Ours+** | **33.9** | **72.1** | **0.60** | **0.89** | **0.138** | **0.84** | **0.19** |

Table 6: Performance gain of our method for joint perception, prediction, and planning.

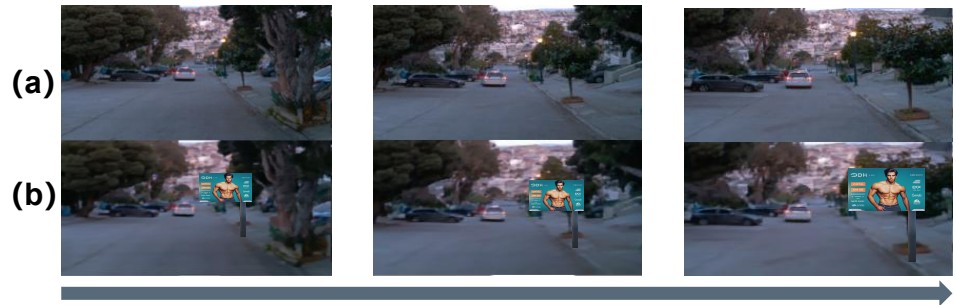

Figure 6: Scene editing. We can insert the new object in the scene, and ensure time consistency.

depth in addressing the issue of camera intrinsic parameters. The utilization of multiple extrinsic parameters helps the algorithm learn the stereo relationship between cameras more effectively.

**Pre-training model.** The 4D reconstruction network is capable of understanding the geometric information of the scene, the motion trajectory of dynamic objects, and the semantic information. To leverage these capabilities for pre-training, we replaced our encoder with the ResNet-50, which is a commonly used base network for many algorithms. We then retrained the 4D reconstruction network on nuScenes dataset, without using any segmentation annotations(without $\mathcal{L}_{sr}$ and $\mathcal{L}_{seg}$). Subsequently, we replaced the encoder of UniAD (Hu et al., 2023) with our pre-trained model and fine-tuned it on the nuScenes dataset. This pre-training processing is fully compliant with VIDAR's protocol, so we copied VIDAR's original results directly. The results, as presented in Table 6, demonstrate that our pre-trained model achieved better performance compared to ViDAR (Yang et al., 2024), highlighting the ability of our algorithm to leverage large-scale unsupervised data for pre-training and improving multiple downstream tasks.

**Scene editing.** The 4D scene reconstruction model enables us to obtain comprehensive 4D geometry information of a scene, which allows for the removal, insertion, and control of objects within the scene. As shown in Figure 6, we added billboards (3D Guassian presentation) to fixed positions in the scene, representing a corner case where cars come to a stop. It is worth mentioning that we can use the existing 3D generation model Tang et al. (2024) to generate any object insertion scene. As can be seen from the figure, the scenario we created exhibits a high level of temporal consistency.

## 5 CONCLUSION

The paper introduces DrivingRecon, a novel 4D Gaussian Reconstruction Model for fast 4D reconstruction of driving scenes using surround-view video inputs. A key innovation is the Prune and Dilate Block (PD-Block), which prunes redundant Gaussian points from adjacent views and dilates points around complex edges, enhancing the reconstruction of dynamic and static objects. Additionally, a dynamic-static rendering approach using optical flow prediction allows for better supervision of moving objects across time sequences. DrivingRecon shows superior performance in scene reconstruction and novel view synthesis compared to existing methods. It is particularly effective for tasks such as model pre-training, vehicle adaptation, and scene editing.

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

# A  MODEL DETAILS

## A.1  PRUNE AND DILATE BLOCK

Below is the PyTorch pseudo-code for the Prune and Dilate Block (PD-Block) presented 1. The pseudo-code outlines the key steps of the PD-Block, including feature concatenation, region partitioning, center proposal, similarity computation, mask generation, and feature aggregation.

The Prune and Dilate Block (PD-Block) begins by computing a value feature and a range view feature from the input feature map. These features are reshaped to accommodate multiple attention heads. If folding is enabled (i.e., fold_w > 1 and fold_h > 1), the feature maps are partitioned into smaller regions to reduce computational overhead.

Next, the block proposes a set of center points evenly distributed in space and computes their corresponding features by averaging the nearest points. A pair-wise cosine similarity matrix between the region features and the center points is calculated and passed through a sigmoid activation after scaling and shifting. A mask is generated based on a threshold to retain significant similarities, ensuring that the most similar points to each center are preserved.

The features are then aggregated by combining the long-term and local features weighted by the mask. Depending on the configuration, the aggregated features can either be returned as center features or dispatched back to each point in the cluster. If regions were previously split, they are merged back into the full feature map. Finally, the output is reshaped to restore the multi-head configuration and projected to produce the final feature map.

## A.2  TEMPORAL CROSS ATTENTION

Due to the inherently sparse nature of multi-view data with minimal overlap, neural networks struggle to accurately capture the geometric information of scenes and objects. To address this limitation, we employ temporal self-attention to integrate temporal features by simultaneously considering both temporal and spatial dimensions (Ren et al., 2024). It is worth emphasizing that we have not made any contribution here, but just copied paper (Ren et al., 2024). These temporal self-attention layers treat the view axis (V) as a separate batch of independent video sequences by transferring the view axis into the batch dimension. After processing, the data is reshaped back to its original configuration, this process looks as:

$$\mathbf{x} = \text{rearrange}(\mathbf{x}, (\text{B T V}) \ \text{H W C} \rightarrow (\text{B V}) \ (\text{T H W}) \ \text{C}) \tag{1}$$

$$\mathbf{x} = \mathbf{x} + \text{TempSelfAttn}(\mathbf{x}) \tag{2}$$

$$\mathbf{x} = \text{rearrange}(\mathbf{x}, (\text{B V}) \ (\text{T H W}) \ \text{C} \rightarrow (\text{B T V}) \ \text{H W C}) \tag{3}$$

where $\mathbf{x}$ is the feature, B H W C are batch size, height, width, and the number of channels. By simultaneously considering temporal and spatial dimensions, temporal self-attention enables neural networks to better capture and interpret the geometric information of scenes and objects, overcoming the limitations caused by sparse view overlaps. Incorporating temporal dynamics enriches the feature maps with contextual information over time, leading to more robust and comprehensive representations of complex scenes.

# B  VISUALIZATION

**Reconstructions, Depth Maps, and Segmentation Maps.** To demonstrate the effectiveness of our algorithm, we randomly selected several examples of scene reconstructions, depth predictions, and segmentation results, as illustrated in Figures 8 and 9. These images reveal that our model consistently achieves high-quality reconstructions across diverse environments, including urban and suburban settings, as well as varying lighting conditions such as day and night. Notably, our method accurately distinguishes between static and moving objects, underscoring its robustness and precision in complex scenes.

**Additional Cases of Novel View Synthesis.** Novel view synthesis is a fundamental capability in scene reconstruction, playing a crucial role in enhancing the generalization performance of downstream tasks. To further validate the effectiveness of our approach, we present additional examples

---

**Algorithm 1** Prune and Dilate Block (PD-Block)

---

**Require:** Input feature map $x \in \mathbb{R}^{B \times C \times W \times H}$
**Ensure:** Output feature map out $\in \mathbb{R}^{B \times C' \times W \times H}$
 1: Compute value features: value $\leftarrow$ self.v$(x)$
 2: Compute range view features: $x \leftarrow$ self.f$(x)$
 3: Rearrange features for multi-head processing:
 4:     $x \leftarrow$ rearrange$(x, $"b (e c) w h $\rightarrow$ (b e) c w h$, e = $ heads)
 5:     value $\leftarrow$ rearrange(value, "b (e c) w h $\rightarrow$ (b e) c w h$, e = $ heads)
 6: **if** fold_w $> 1$ **and** fold_h $> 1$ **then**
 7:     Get current shape: $(b_0, c_0, w_0, h_0) \leftarrow x$.shape
 8:     Assert feature map is foldable:
 9:         **assert** $w_0$ mod fold_w $= 0$ **and** $h_0$ mod fold_h $= 0$
 10:     Fold feature maps:
 11:         $x \leftarrow$ rearrange$(x, $"b c (f1 w) (f2 h) $\rightarrow$ (b f1 f2) c w h,
 12:         $f1 = $ fold_w$, f2 = $ fold_h)
 13:         value $\leftarrow$ rearrange(value, "b c (f1 w) (f2 h) $\rightarrow$ (b f1 f2) c w h,
 14:         $f1 = $ fold_w$, f2 = $ fold_h)
 15: **end if**
 16: Propose centers: centers $\leftarrow$ self.centers_proposal$(x)$
 17: Compute center features:
 18:     value_centers $\leftarrow$ rearrange(self.centers_proposal(value),
 19:         "b c w h $\rightarrow$ b (w h) c")
 20: Compute pair-wise cosine similarity:
 21:     sim $\leftarrow \sigma$ (self.sim_beta + self.sim_alpha $\cdot$ pairwise_cos_sim(
 22:         centers.reshape$(b, c, -1).permute(0, 2, 1)$,
 23:         x.reshape$(b, c, -1).permute(0, 2, 1))$ )
 24: Generate mask:
 25:     $($sim_max$, $sim_max_idx$) \leftarrow$ sim.max$($dim $= 1, $keepdim $=$ True$)$
 26:     mask $\leftarrow$ zeros_like(sim)
 27:     mask.scatter_$(1, $sim_max_idx$, 1.)$
 28:     sim $\leftarrow$ sim $\times$ mask
 29: Rearrange value for aggregation: value2 $\leftarrow$ rearrange(value, "b c w h $\rightarrow$ b (w h) c")
 30: Aggregate features:
 31:     out $\leftarrow \frac{(\text{value2.unsqueeze}(1) \times \text{sim.unsqueeze}(-1)).\text{sum}(\text{dim}=2) + \text{value\_centers}}{\text{sim.sum}(\text{dim}=-1, \text{keepdim}=\text{True}) + 1.0}$
 32: **if** self.return_center **then**
 33:     Rearrange output to center format:
 34:         out $\leftarrow$ rearrange(out, "b (w h) c $\rightarrow$ b c w h$, w = ww)$
 35: **else**
 36:     Dispatch features to each point:
 37:         out $\leftarrow$ (out.unsqueeze(2) $\times$ sim.unsqueeze$(-1)).$sum(dim $= 1)$
 38:         out $\leftarrow$ rearrange(out, "b (w h) c $\rightarrow$ b c w h$, w = w)$
 39: **end if**
 40: **if** fold_w $> 1$ **and** fold_h $> 1$ **then**
 41:     Merge folded regions back:
 42:         out $\leftarrow$ rearrange(out, "(b f1 f2) c w h $\rightarrow$ b c (f1 w) (f2 h),
 43:         $f1 = $ fold_w$, f2 = $ fold_h)
 44: **end if**
 45: Rearrange back to multi-head format:
 46:     out $\leftarrow$ rearrange(out, "(b e) c w h $\rightarrow$ b (e c) w h$, e = $ heads)
 47: Project output: out $\leftarrow$ self.proj(out)
 48: **return** out

---

of novel view renderings in Figures 10 and 11. The high quality of these synthesized views demonstrates the efficacy of our method in generating realistic and coherent scene perspectives from new viewpoints.

## C  ABLATION EXPERIMENT

Our framework incorporates several critical hyperparameters that are pivotal to the model's performance. Specifically, depth supervision ($\lambda_c$), 3D positional encoding regularization ($\lambda_{PE}$), and segmentation loss weighting ($\lambda_{seg}$) are identified as the three most influential hyperparameters in this study. To evaluate their effects, we conducted extensive ablation experiments, the results of which are presented in Figure 7.

The results reveals that all forms of regular supervision contribute positively to the model's performance. In particular, depth supervision ($\lambda_c$) significantly enhances reconstruction quality compared to scenarios without additional supervision. Conversely, increasing the weight of segmentation supervision ($\lambda_{seg}$) leads to a decrease in reconstruction performance. This adverse effect is attributed to the introduction of noise during the segmentation supervision phase, which degrades the model's performance.

Based on the evaluation protocol outlined in Table 1, we compared the speed and PSNR of our method against traditional optimization methods as shown in Tab. 7:

| Method | PSNR | SSIM | LPIPS | Time Cost |
|---|---|---|---|---|
| 3D-GS | 24.91 | 0.71 | 0.16 | 5.5h |
| DrivingGaussian | 26.12 | 0.74 | 0.13 | 6.2h |
| **Ours** | **23.70** | **0.68** | **0.17** | **1.21s** |

Table 7: Comparison of our method with traditional optimization methods.

As indicated in the table, our algorithm performs comparably to traditional optimization methods in terms of PSNR while significantly reducing time costs. This efficiency makes our method more suitable for data-driven applications, such as driving simulators.

In addition to optimization-based methods, we further evaluated the efficiency of other SOTA forward generalizable models.

| Method | PSNR | SSIM | LPIPS | Time Cost | Memory |
|---|---|---|---|---|---|
| LGM | 19.52 | 0.52 | 0.32 | 1.82s | 21.42G |
| pixelSplat | 20.54 | 0.58 | 0.28 | 2.44s | 19.65G |
| MVSplat | 21.33 | 0.64 | 0.24 | 1.64s | 15.47G |
| L4GM | 20.01 | 0.54 | 0.30 | 1.98s | 23.74G |
| Ours | 23.70 | 0.68 | 0.17 | 1.21s | 11.08G |

Table 8: The efficiency comparison of SOTA methods.

As shown in the table, our method is significantly optimal in reasoning speed and memory usage. For the automatic driving scene, the efficiency of our method is due to: (1) Multi-view fusion better integrates multiple views with small overlap through the form of range view. (2) Timing fusion is the fusion of highly compressed implicit features, which greatly reduces memory and inference delay. (3) Image encoder and decoder are shared for different perspectives and can be inferred in parallel.

Disadvantages of other methods: (1) For the input of multiple graphs, MVSplat (Chen et al., 2024) needs to calculate the cost volume between any two images, which greatly increases the computational memory and inference delay. (2) LGM (Tang et al., 2024) and L4GM (Ren et al., 2024) cat all the images into a multi-view attention fusion network. The uncompressed image sent to the view fusion network consumes memory and increases inference delay. In addition, the small overlap of different perspectives in the driving scene does not require such redundant attention mechanisms. (3) pixelSplat (Charatan et al., 2023) uses the polar coordinate attention fusion mechanism to integrate different perspectives. The small overlap of different perspectives in the driving scene does not require such redundant attention mechanisms. Specifically, a large number of queries are empty.

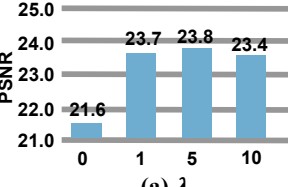 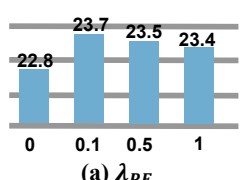 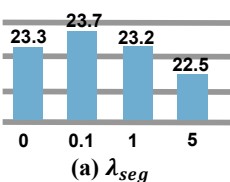

Figure 7: Ablation study of hyperparameters. $\lambda_c, \lambda_{PE}, \lambda_c$ is the supervision weight of the depth supervision, 3D-PE regular and segmentation.

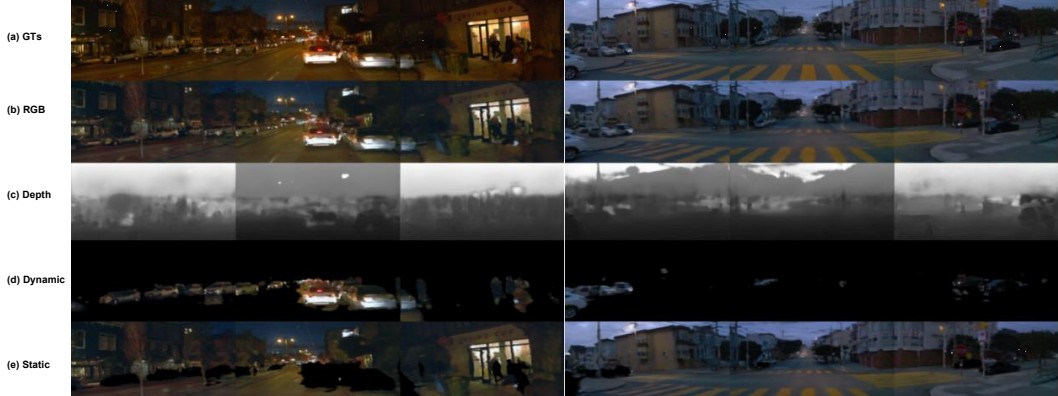

Figure 8: Reconstructed visualization: (a) ground truth, (b) Reconstructed rgb images, (c) Depth maps, (d) dynamic object reconstruction, and (e) static object reconstruction **(zoom in for the best view.)**

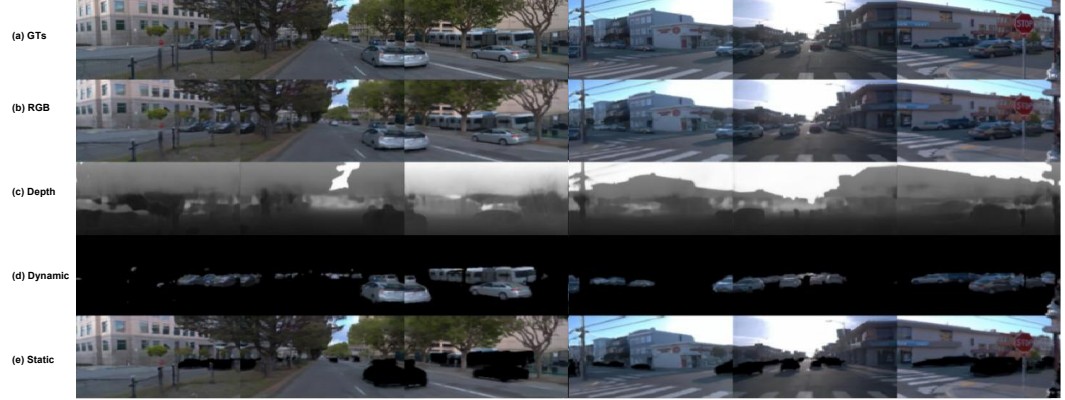

Figure 9: Reconstructed visualization: (a) ground truth, (b) Reconstructed rgb images, (c) Depth maps, (d) dynamic object reconstruction, and (e) static object reconstruction **(zoom in for the best view.)**

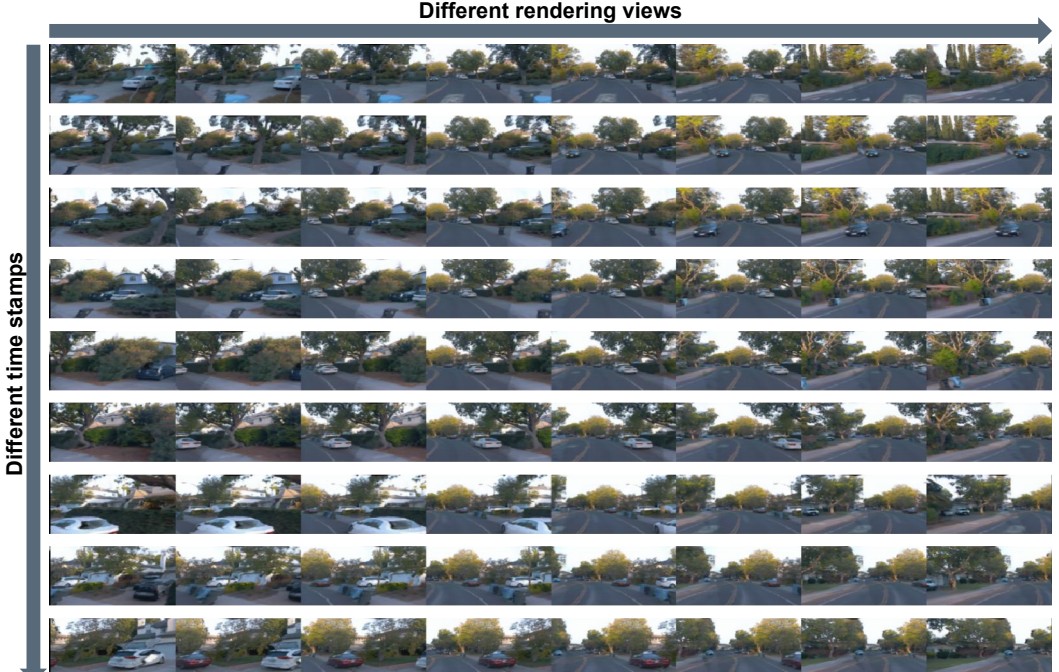

Figure 10: Novel view rendering. Based on the predicted Gaussians, we render different views at different times. The novel views are of very high quality and very high spatio-temporal consistency **(zoom in for the best view.)**

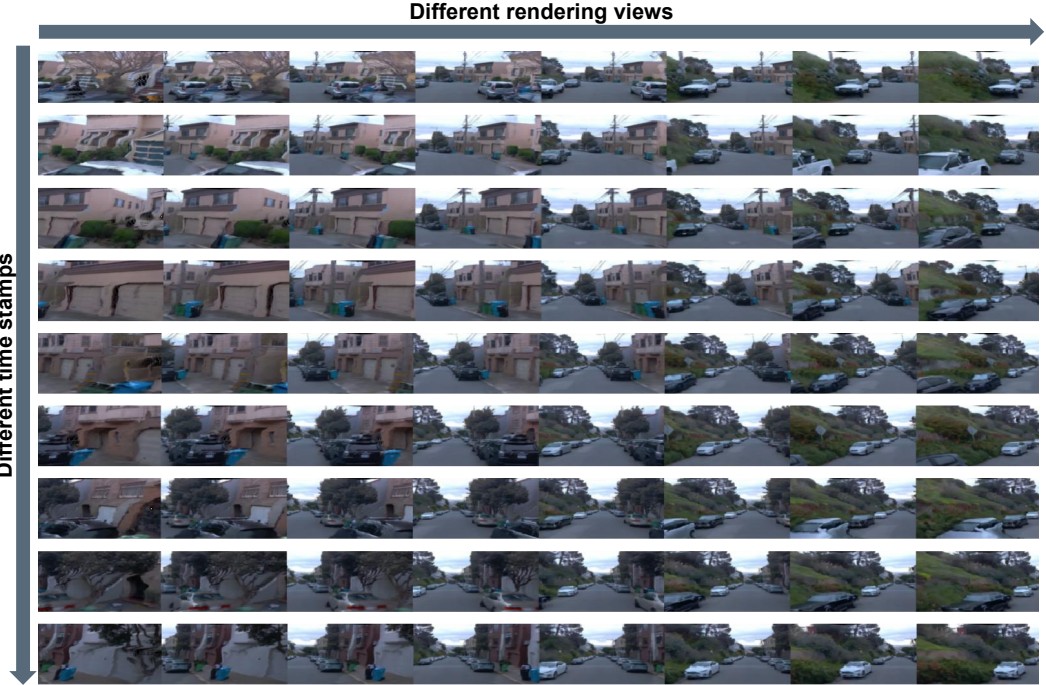

Figure 11: Novel view rendering. Based on the predicted Gaussians, we render different views at different times. The novel views are of very high quality and very high spatio-temporal consistency **(zoom in for the best view.)**

