# OpenReview forum: "DrivingRecon: Large 4D Gaussian Reconstruction Model For Autonomous Driving"
_ICLR.cc/2025/Conference — Submitted to ICLR 2025_

### Official Review · Reviewer_EbAv · 2024-10-22

**Soundness:** 3
**Presentation:** 2
**Contribution:** 2
**Rating:** 5
**Confidence:** 3

**Summary:**

This paper proposes a feed-forward 4D reconstruction method that generates 4D scenes from surround-view video inputs in a single feed-forward pass.
The method involves 3D Position Encoding, Temporal Cross Attention, Gaussian Adapter, and Prune and Dilate Block. All these modules consist of the feed-forward 4D reconstruction pipeline.
The PD-Block learns to prune redundant Gaussian points from different views and background regions and dilate Gaussian points for complex objects, enhancing the quality of reconstruction.
This paper also presents rendering strategies for both static and dynamic components, enabling efficient supervision of rendered images across temporal sequences.

**Strengths:**

1.This paper first explores a feed-forward 4D reconstruction method for surround-view driving scenes, which promotes the development of feed-forward technology in the field of 4D reconstruction.

2.The proposed PD-Block learns to prune and dilate the Gaussian points and allows for Gaussian points that are not strictly pixel-aligned, which is innovative.

**Weaknesses:**

1.The training process requires depth map ground truth, whereas comparison methods like Pixelsplat and MVSpalt can be trained without it. This reliance on depth ground truth during training restricts its practical applicability.

2.The dynamic objects are decomposed through segmentation and have only few categories (vehicles and people). This approach only separates dynamic and static pixels based on semantics, limiting its ability to achieve comprehensive 4D reconstruction of all dynamic objects.

3.Compared to scene-optimized methods, feed-forward reconstruction provides the advantage of generalization, eliminating the need of test-time optimization for each new scene (though it may lead to some decrease in accuracy compared to the scene-optimized method). In the papers of comparing methods MVSplat and PixelSplat, both of them present running time and memory consumption, demonstrating the efficiency of their feed-forward approaches. However, in this paper, while the authors claim their method is feed-forward, they do not provide an analysis of its running time and memory usage. I recommend including this efficiency analysis and comparing it with other methods to strengthen the evaluation.

Besides, if the authors believe that efficiency is not a concern of this paper, then comparisons with other offline scene-optimized methods (e.g., DrivingGaussian) should be included.

4.If the possible application is to develop real-world simulators in autonomous driving (mentioned in the abstract of the paper), then there is no high requirement for the efficiency of reconstruction, and the existing offline scene-optimized 4D reconstruction method is also acceptable. However, feed-forward does not seem to have an advantage in terms of reconstruction accuracy.

**Questions:**

1.What does “DA-Block” in line 202 refer to? It is not mentioned in the context.

2.Please refer to the questions and suggestions in the Weaknesses part.

---

> ### Author Response · Authors · 2024-11-22
> **Author Response to Reviewer EbAv (Part 1)**
>
> ---
>
> ### **（W1）The Need for Depth Map Ground Truth**
>
> We appreciate the reviewer's insights on the necessity of depth supervision. While depth supervision is not strictly required in our framework, we acknowledge that it can enhance the speed of model convergence and lead to improved final results.
>
> To further clarify our findings, we conducted a comparative experiment to evaluate the impact of depth supervision under the experimental conditions detailed in Table 1. The results are presented below:
>
> |  | PSNR | SSIM | LPIPS |
> | --- | --- | --- | --- |
> | Ours w/o depth | 22.23 | 0.63 | 0.24 |
> | Ours | 23.70 | 0.68 | 0.17 |
>
> The table indicates that our algorithm can still learn geometric information through temporal consistency, even in the absence of depth constraints. Furthermore, in driving scenarios, depth information from LiDAR is typically readily available, as discussed in other works (e.g., paper [1]).
>
> ### **（W2）The Limitations of Semantic Segmentation**
>
> Although we now only segment cars, two-wheelers and pedestrians as dynamic objects. Here are some simple ideas for segmenting all dynamic objects. For instance, calculating optical flow from videos could help identify points in dynamic regions, which could then serve as prompts for a segmentation algorithm like SAM. This would enable us to segment a broader range of dynamic objects.
>
> We believe that this limitation should not overshadow the value of our paper, as many advanced and well-regarded driving reconstruction algorithms share this same challenge [1, 2, 3, 4]. Addressing this issue is a minor aspect of our framework and does not reflect the core innovations presented in our work.
>
> [1] Drivinggaussian: Composite gaussian splatting for surrounding dynamic autonomous driving scenes. CVPR2024.
>
> [2] Holistic urban 3d scene understanding via gaussian splatting. *CVPR2024*
>
> [3] OmniRe: Omni Urban Scene Reconstruction." *arXiv preprint arXiv:2408.16760* (2024).
>
> [4] Street gaussians for modeling dynamic urban scenes. ECCV2025
>
> ### **（W3）Lack of Efficiency Analysis**
>
> We appreciate the reviewer pointing out the need for an efficiency analysis. We will include an analysis of the efficiency of our method in the paper. According to the evaluation protocol in Table 1, we compared the speed and PSNR of our method against traditional optimization methods:
>
> |  | PSNR | SSIM | LPIPS | Time Cost |
> | --- | --- | --- | --- | --- |
> | 3D-GS | 24.91 | 0.71 | 0.16 | 5.5h |
> | DrivingGaussian | 26.12 | 0.74 | 0.13 | 6.2h |
> | Ours | 23.70 | 0.68 | 0.17 | 1.21s |
>
> As shown in the table, our algorithm performs comparably to traditional optimization methods in terms of PSNR, while significantly reducing time cost.  The experiment was updated to part C of supplementary materials. To prove the effectiveness of our reconstruction, we have provided some reconstruction videos where you can observe our method at the following link: https://anonymize58426.github.io/Drive-Recon/. In the videos, you can see the lane lines being translated between 3 to 12 seconds, and the viewpoint being rotated between 17 to 26 seconds. This demonstrates that the scenes reconstructed by our algorithm maintain geometric consistency.

---

> ### Author Response · Authors · 2024-11-22
> **Author Response to Reviewer EbAv (Part 2)**
>
> ### **（W4）The Necessity of a 4D Feedforward Network**
>
> While we acknowledge that the rendering quality of the feedforward network in driving scenes may currently be slightly lower than that of optimization methods, we are confident that feedforward networks will surpass optimization methods in the near future, especially with advancements in generative techniques. We would like to highlight several key points regarding the necessity of feedforward networks:
>
> 1. **General Driving Pre-Training Model**: The 4D feedforward network serves as a general pre-training model for driving tasks. By learning 4D reconstruction, the network extracts geometric information from the entire scene while simultaneously predicting motion information for moving objects. This pre-training approach captures more geometric and temporal features than existing methods, as verified in Table 6 of our paper. Our model can serve as a pre-training tool to enhance the performance of driving algorithms in multiple tasks such as perception and planning without relying on labels like segmentation or 3D bounding boxes.
> 2. **Possibility of a 4D World Model**: Current driving world models often exist in video form without explicit geometric representation, leading to limitations in perspective and temporal consistency. Upgrading our reconstruction model to support predictable 4D generation could address these issues. A 4D world model could provide explicit geometry to constrain end-to-end planning, representing a significant advancement for the field.
> 3. **Scaling Capabilities**: The 4D feedforward network has the potential for scalability. It can be trained using an abundance of driving videos available on the Internet. This extensive training could greatly enhance reconstruction capabilities, which can then be leveraged for tasks such as perception and planning, either through pre-training or as part of the world model.
> 4. **Importance of Generalization and Efficiency**: Building a 4D scene for a new city using optimization methods is very time-consuming, particularly with long videos capturing thousands of scenarios. For example, a 200-frame video (10 seconds) might take around 6 hours for an optimization-based reconstruction method, which is not feasible in practice.
>
> ### **（Q）Minor Typos**
>
> We would like to thank the reviewer for catching the typo regarding “DA-Block” in line 202; it should indeed be “PD-Block.” We will proofread the paper thoroughly to enhance its writing, presentation, and layout.
>
> Thank you for your attention to our responses. If you have any further questions or if anything remains unclear, please don’t hesitate to let us know. We would be more than happy to discuss your concerns in greater detail.

---

> ### Author Response · Authors · 2024-11-25
> **Discussion request**
>
> Dear Reviewer EbAv,
>
> I would like to express my sincere gratitude to you for your constructive comments. As the ICLR discussion phase is almost over, I wanted to kindly ask if there are any remaining questions or clarifications needed regarding our responses. Please feel free to reach out at any time.
>
> We would be truly grateful if you could consider raising our rating, as your support is crucial for the potential acceptance of our work.
>
> Best wishes,
>
> Authors

---

> > ### Comment · Reviewer_EbAv · 2024-11-25
> >
> > I appreciate the authors' effort in pioneering the exploration of feed-forward autonomous 4DGS reconstruction and the authors’ rebuttal does solve some of my concerns (W1&W2).
> >
> > However, I think the performances presented in this paper may not be sufficient to fully support its motivation. This paper requires multiple inputs (depth supervision/semantic segmentation) and heavy network architecture(image encoder&decoder/TCA), but shows a significant drop in PSNR compared to optimization based reconstruction method(e.g. drivingGaussian), and does not provide efficiency comparisons with other feed-forward reconstruction(As the proposed network is more complex than other methods, I guess it will need more time or memory). The aforementioned problems limit the pratical use of this paper.
> >
> >
> > More comments
> >
> > W3: The original 3DGS is designed for static scene without depth or segmentation supervision which makes comparison to 3DGS less meaningful.
> >
> > W4-4: The authors claim that a 200-frame video might take around 6h for optimization-based reconstruction, while I believe the time overhead should not be this significant.

---

> ### Author Response · Authors · 2024-11-25
> **Further Response**
>
> Dear Reviewer EbAv,
>
> Thank you for carefully reading our response and providing further comments. I hope our further discussion will allow you to change your opinion.
>
> Most importantly, multiple supervision and complex modules are not a weakness to reject a paper: (1) optimization-based driving reconstruction algorithm approaches will also have multiple supervision (point cloud initialization and segmentation of dynamic and static objects) [1,2,3,4]. When reasoning, our models do not need depth and segmentation labels.  Besides, our method can indeed be trained without requiring semantic segmentation or 3D boxes. As shown in Table 4(a), we can learn geometry through perspective consistency and point cloud depth information without the need for Dynamic and Static Rendering (DS-R). In the W1 response we also explained that our approach can be done without depth supervision. (2) For learnable driving tasks, such as perception task, they also have multiple modules including temporal fusion model, multi-view fusion model, image encoder model, image decoder and detection [5, 6, 7].
>
> [1] Drivinggaussian: Composite gaussian splatting for surrounding dynamic autonomous driving scenes. *CVPR 2024*
>
> [2] OmniRe: Omni Urban Scene Reconstruction. *arXiv preprint arXiv:2408.16760* (2024).
>
> [3] Street gaussians for modeling dynamic urban scenes. ECCV2025
>
> [4] Unisim: A neural closed-loop sensor simulator. CVPR2023.
>
> [5] Exploring object-centric temporal modeling for efficient multi-view 3d object detection. CVPR2023
>
> [6] Bevformer: Learning bird’s-eye-view representation from multi-camera images via
> spatiotemporal transformers. ECCV2023
>
> [7] Panoocc: Unified occupancy representation for camera-based 3d panoptic segmentation. CVPR2024
>
> ### **Compare the existing FeedForword method**
>
> Thanks to the reviewer's reminder, we should indeed evaluate the efficiency of other SOTA forward generalizable models.
>
> |  | PSNR | SSIM | LPIPS | Time Cost | Memory |
> | --- | --- | --- | --- | --- | --- |
> | LGM | 19.52 | 0.52 | 0.32 | 1.82s | 21.42G |
> | pixelSplat | 20.54 | 0.58 | 0.28 | 2.44s | 19.65G |
> | MVSplat | 21.33 | 0.64 | 0.24 | 1.64s | 15.47G |
> | L4GM | 20.01 | 0.54 | 0.30 | 1.98s | 23.74G |
> | Ours | 23.70 | 0.68 | 0.17 | 1.21s | 11.08G |
>
> As shown in the table, our method is significantly optimal in reasoning speed and memory usage. For the automatic driving scene, the efficiency of our method is due to: (1) Multi-view fusion better integrates multiple views with small overlap through **the form of range view**. (2) Temporal fusion is the fusion of highly compressed implicit features, which greatly reduces memory and inference delay. (3) Image encoder and decoder are **shared** for different views and can be inferred in parallel.
>
> Disadvantages of other methods: (1) For the input of multiple views, MVSplat needs to calculate the cost volume between any two image pairs, which greatly increases the computational memory and inference delay. (2) LGM and L4GM cat all the images into a multi-view attention fusion network. The uncompressed image sent to the view fusion network consumes memory and increases inference delay. In addition, the small overlap of different perspectives in the driving scene does not require such redundant attention mechanisms. (3) pixelSplat uses the polar coordinate attention fusion mechanism to integrate different perspectives. The small overlap of different perspectives in the driving scene does not require such redundant attention mechanisms. Specifically, a large number of queries are empty.
>
> ### **Limitations of the Original 3DGS**
> In our efficiency comparison experiments, the original 3DGS heavily also relies on point cloud supervision to initialize the Guassian points. At each time step $t$, a 3DGS model needs to be trained. This is why it takes a significant amount of time, approximately 5.5 hours, to reconstruct a 200-frame video using 3DGS. However, the reconstruction performance tends to overfit, and its ability to synthesize novel view is bad, as shown in the table below.
>
> |  | Reconstruction |  |  | Novel View |  |  |  |
> | --- | --- | --- | --- | --- | --- | --- | --- |
> |  | PSNR | SSIM | LPIPS | PSNR | SSIM | LPIPS | Time Cost |
> | 3D-GS | 24.91 | 0.71 | 0.16 | 18.81 | 0.55 | 0.31 | 5.5h |
> | DrivingGaussian | 26.12 | 0.74 | 0.13 | 22.34 | 0.74 | 0.19 | 6.2h |
> | Ours | 23.70 | 0.68 | 0.17 | 20.63 | 0.61 | 0.21 | 1.21s |
>
> As shown in the table, 3DGS significantly deteriorated in new view synthesis. At each time step, there are only a few observed viewpoints in driving scenes (such as 6 views in nuScenes and 5 in Waymo) to supervise 3DGS. Besides, the overlap between these surround views is minimal, making it challenging for 3DGS to accurately learn the geometry. To address this, DrivingGaussian and ours use segmentation to identify the static background and perform cross-temporal supervision.
>
>  If you have any further questions or if anything remains unclear, please don’t hesitate to let us know.

---

> > ### Comment · Reviewer_conn · 2024-12-01
> >
> > Thanks the author for the response. I move the discussion here as I share similar concerns as Reviewer EbAv.
> > 1. The quality in the video is way too low to call it a reconstruction method.
> > 2. While the papers you mentioned can be furthur simplified, I still feeling drivingRecon is more compliciated and entangled than necessary, with the requirement of more specialized modules, pretrained models and prior.
> > 3. The speed comparison is not solid. In my experience, Streetgaussian, with some hyperparameter tuned,  trained for 5-10 minutes could end up with  higher results than DrivingRecon. And the claim of DrivingRecon's 1.21s timecost is not full log reconstruction cost (even though the authors don't mentioned, but I expect it's per timestamp). So this seems an unfair (at least unsolid) comparison.  Especially, the comparison setup for 3DGS ( "At each time step , a 3DGS model needs to be trained. ") is particularly problematic, as these methods can be trained on all frames simultaneously to achieve better and more temporally consistent results.   Lastly, DrivingRecon requires significant GPU resources for training (24 NVIDIA A100 80GB GPUs as stated in the paper).  With these training cost, we could reconstruct hundreds of logs in higher quality.

---

> > > ### Author Response · Authors · 2024-12-03
> > > **Further Response**
> > >
> > > Dear Reviewers,
> > >
> > > Thanks the author for the response. I always admit that optimized methods are still of better quality than generalized methods. I'm not saying that my model can be simplified, I'm saying that I can make better use of some new techniques to further optimize the algorithm. Five minutes to train Streetgaussian still takes more than 200 times longer than generalizable methods. And the iterative optimization method does not require less memory than our algorithmic inference needs. By the way, using the original Streetgaussian parameters required 2 hours of training on the 3090 or A100 and 1.5 hours on the 4090. The original 3DGS cannot be trained across time series because it has no separation of static and moving objects. And I'm using DrivingGaussian instead of Streetgaussian. I insist that generalizable Gauss is promising, and you are too harsh on the first 4D driving generalizable reconstruction paper.
> > >
> > > Best,
> > > Authors

---

### Official Review · Reviewer_3kKi · 2024-11-01

**Soundness:** 3
**Presentation:** 2
**Contribution:** 3
**Rating:** 6
**Confidence:** 4

**Summary:**

This paper proposes a learning-based reconstruction method in a feed-forward manner in driving scenarios. It could predict 4D Gaussian primitives from multi-view temporal input. It is a very early work that explores learning-based generalizable reconstruction and rendering for autonomous driving. This paper also introduces a couple of downstream applications such as model pre-training and vehicle adaptation.

**Strengths:**

-  It is a very early work that explores learning-based generalizable reconstruction methods for autonomous driving, demonstrating this paradigm could work in real-world driving scenarios.
- This paper is comprehensive since it not only develops the methods but also incorporates potential applications such as perception and driving tasks.
- The self-supervised pretraining task is insightful.

**Weaknesses:**

- This paper does not demonstrate the model's generalization to different viewpoints. The authors claim the ability of vehicle adaption. However, only the camera intrinsic is changed. Could the predicted 4D Gaussians produce good rendering quality in viewpoints beyond the driving trajectories (different extrinsic)? A recent work[1] explores this direction.

- The resolution is relatively low.  The produced rendering quality cannot meet the requirements of practical use, such as camera simulation.

- It would be better to show the inference latency.

- The authors do not provide video demonstrations of the rendering results. It is hard to have a intuitive understanding of the actual performance.

[1] Freevs: generative view synthesis on free driving trajectory.

**Questions:**

How does the scene edit (Fig.6) work? This procedure can be more detailed.

---

> ### Author Response · Authors · 2024-11-22
> **Author Response to Reviewer 3kKi**
>
> Thank you for your thoughtful comments and for taking the time to review our work. Your feedback is genuinely appreciated, and I hope to clarify and enhance our responses to your concerns.
>
> ### **(W1 and W4) Generalization to Different Viewpoints**
>
> To demonstrate the effectiveness of our method in generating new perspectives, we have provided a series of reconstruction videos available at the following link: [Drive-Recon Videos](https://anonymize58426.github.io/Drive-Recon/). In these videos, you can observe the translation of lane lines between 3 to 12 seconds and the rotation of viewpoints between 17 to 26 seconds. This effectively illustrates that the scenes reconstructed by our algorithm maintain geometric consistency across varying perspectives.
>
> Additionally, I appreciate your reference to the Freevs method; it is indeed an intriguing approach. I believe we can adapt some of its useful components to enhance our model's ability to generate new viewpoints.
>
> ### **(W2) Limitations of Resolution**
>
> Currently, it seems that feedforward networks yield slightly lower rendering quality in driving scenarios compared to traditional optimization methods. However, I am hopeful that advancements in network architectures, training strategies, and generative techniques will soon enable feedforward networks to exceed the capabilities of optimization methods. Our paper serves as a preliminary study in the driving domain and provides a codebase that may accelerate progress in this area.
>
> Moreover, existing state-of-the-art generalizable Gaussian splatting algorithms tend to operate at lower resolutions and realism, particularly in indoor scenes [1, 2, 3]. Therefore, we believe our work is still at the forefront of the field of generalizable Gaussian splatting.
>
> References:
>
> 1. Mvsplat: Efficient 3D Gaussian Splatting from Sparse Multi-View Images. *ECCV* 2025.
> 2. Large Spatial Model: End-to-End Unposed Images to Semantic 3D. NeurIPS 2024.
> 3. FreeSplat: Generalizable 3D Gaussian Splatting Towards Free-View Synthesis of Indoor Scene Reconstruction. NeurIPS 2024.
>
> ### **(W3) Lack of Efficiency Analysis**
>
> Thank you for highlighting the need for an efficiency analysis. We will include a thorough evaluation of our method's efficiency in the revised paper. Based on the evaluation protocol outlined in Table 1, we compared the speed and PSNR of our method against traditional optimization methods:
>
> | Method | PSNR | SSIM | LPIPS | Time Cost |
> | --- | --- | --- | --- | --- |
> | 3D-GS | 24.91 | 0.71 | 0.16 | 5.5h |
> | DrivingGaussian | 26.12 | 0.74 | 0.13 | 6.2h |
> | **Ours** | **23.70** | **0.68** | **0.17** | **1.21s** |
>
> As indicated in the table, our algorithm performs comparably to traditional optimization methods in terms of PSNR while significantly reducing time costs. This efficiency makes our method more suitable for data-driven applications, such as driving simulators. The experiment was updated to part C of supplementary materials.
>
> ### **(Q) Details of Scene Editing**
>
> To edit scenes, we can utilize existing 3D generation models to create Gaussian representations of arbitrary objects, such as LGM [1]. We then modify the Gaussian representation of the x, y, and z positions (in world coordinates) to place them appropriately within the driving scenario. Importantly, the Gaussians predicted by our method are represented within the world coordinate system.
>
> Reference:
>
> 1. LGM: Large Multi-View Gaussian Model for High-Resolution 3D Content Creation.
>
> Thank you for your attention to our responses. If you have any further questions or if anything remains unclear, please don’t hesitate to let us know. We would be more than happy to discuss your concerns in greater detail.

---

> ### Author Response · Authors · 2024-11-25
> **Discussion request**
>
> Dear Reviewer 3kKi,
>
> I would like to express my sincere gratitude to you for your constructive comments. As the ICLR discussion phase is almost over, I wanted to kindly ask if there are any remaining questions or clarifications needed regarding our responses. Please feel free to reach out at any time.
>
> We would be truly grateful if you could consider raising our rating, as your support is crucial for the potential acceptance of our work.
>
> Best wishes,
>
> Authors

---

> > ### Comment · Reviewer_3kKi · 2024-11-25
> > **Thanks for the response**
> >
> > Thanks for providing the additional experiment. However, I asked for the inference latency (rendering ) in my original review. The authors seem to provide the optimization time for 3DGS and DrivingGaussian.
> >
> > The generalization of new viewpoints seems to have some issues since there are clear unreasonable deformations or artifacts when laterally shifting the viewpoints.

---

> ### Author Response · Authors · 2024-11-25
> **Further Response**
>
> Dear 3kKi
>
> Thank you for carefully reading our response and providing further comments. Here is our further response:
>
> ### **Compare the existing FeedForword method**
>
> We apologize for our misunderstanding. Our method in the table is inference latency, and the other optimization method is optimization time. The  inference latency of **rendering part** is only about 0.04 seconds. In addition to optimization-based methods, we further evaluated the efficiency of other SOTA forward generalizable models.
>
> |  | PSNR | SSIM | LPIPS | Time Cost | Memory |
> | --- | --- | --- | --- | --- | --- |
> | LGM | 19.52 | 0.52 | 0.32 | 1.82s | 21.42G |
> | pixelSplat | 20.54 | 0.58 | 0.28 | 2.44s | 19.65G |
> | MVSplat | 21.33 | 0.64 | 0.24 | 1.64s | 15.47G |
> | L4GM | 20.01 | 0.54 | 0.30 | 1.98s | 23.74G |
> | Ours | 23.70 | 0.68 | 0.17 | 1.21s | 11.08G |
>
> As shown in the table, our method is significantly optimal in reasoning speed and memory usage. For the automatic driving scene, the efficiency of our method is due to: (1) Multi-view fusion better integrates multiple views with small overlap through the form of range view. (2) Timing fusion is the fusion of highly compressed implicit features, which greatly reduces memory and inference delay. (3) Image encoder and decoder are shared for different perspectives and can be inferred in parallel.
>
> Disadvantages of other methods: (1) For the input of multiple graphs, MVSplat needs to calculate the cost volume between any two images, which greatly increases the computational memory and inference delay. (2) LGM and L4GM cat all the images into a multi-view attention fusion network. The uncompressed image sent to the view fusion network consumes memory and increases inference delay. In addition, the small overlap of different perspectives in the driving scene does not require such redundant attention mechanisms. (3) pixelSplat uses the polar coordinate attention fusion mechanism to integrate different perspectives. The small overlap of different perspectives in the driving scene does not require such redundant attention mechanisms. Specifically, a large number of queries are empty.
>
> ### **Some issues of novel views**
>
> For driving scenes, synthesizing new views is still a very challenging problem. Even well-known optimization methods, such as DrivingGuassian, do not render new perspectives well. The Freevs you mentioned is a good new view synthesis solution, even if it is only the latest paper appearing on arxiv. In addition, Freevs is only a generative method for synthesizing new view images, rather than a 3D/4D reconstruction method for predicting Gaussians. These are the two routes. DriveRecon makes it very easy to incorporate these new method into a synthetic solution. I believe our approach has great potential.
>
> Thank you for your attention to our responses. If you have any further questions or if anything remains unclear, please don’t hesitate to let us know. We would be more than happy to discuss your concerns in greater detail.

---

### Official Review · Reviewer_conn · 2024-11-03

**Soundness:** 3
**Presentation:** 3
**Contribution:** 2
**Rating:** 5
**Confidence:** 4

**Summary:**

Unlike previous methods (e.g., 3DGS/NeRF) that require thousands of iterations to reconstruct a scene, this work aims to predict a 3D scene representation using a neural network.

 The authors make several design choices to make this pipeline work (PD-block, regularization, 3D encoding, etc.).

Experiments conducted on Waymo demonstrate better performance compared to other generalizable approaches.

**Strengths:**

1. A generalizable and scalable approach that allows training of large models to learn priors from extensive data, generalizing to novel scenes.
2. **Almost** no 3D bounding box labels required for dynamic scenes, enhancing scalability.
3. Detailed explanations and extensive experiments on cross-data evaluation, downstream applications (data augmentation, pretrained model for perception, scene editing).

**Weaknesses:**

1. Overcomplicated design:
   While I appreciate the effort in developing a generalizable model with dynamic-static decomposition, the model seems quite complex, requiring:
   * Multiple modules (image encoder-decoder, temporal cross-attention, Gaussian adapter, PD block, etc.)
   * Numerous regularization terms
   * Several pretrained models (DepthNet, DeepLab, SAM)

   This complexity may hinder downstream applications when used as a pretrained model. For instance, how fast is the model? Is it efficient enough for use in autonomy systems?

2. The realism is still lower compared to optimization-based approaches (e.g., 3DGS), and can only operate on low resolution (256x512) with a limited number of images.

3. (Minor point) The writing seems somewhat rushed, lacking thorough proofreading. Some potential issues:
   * L155, "corresponding intrinsic parameter E" should be K
   * L414 "evaluation on NOTA-DS6" should be Diversity-54

**Questions:**

**Regarding efficiency and comparison with 3DGS**

What is the computational cost to train the model (how many hours on 24 GPUs)?
How long does it take to reconstruct a 3D scene representation using your approach during inference? How does the efficiency compare to 3DGS, e.g., StreetGaussian on 256x512?

How does the realism compare to 3DGS (e.g., StreetGaussian at 256 × 512)? It's okay if it's worse; I'm just curious.



**On 3D labels**
What is the performance without using 3D bounding boxes at all? I note that you use 3D bounding boxes as prompts for SAM. A label-free approach would make this work more impactful.

**On downstream applications**
How is UniAD implemented in Waymo? Would it be possible to conduct your experiments on nuScenes to follow the setting/implementation of UniAD?

**Miscellaneous**:
* How many frames are in the input during training?
* In Table 4b, what does "Training Num" refer to? Do you mean number of scenes? The PSNR seems quite high compared to Table 3.

Some questions may require additional experiments; please disregard if they're not feasible. However, I'm particularly interested in the efficiency and comparison with 3DGS.

---

> ### Author Response · Authors · 2024-11-22
> **Author Response to Reviewer conn (Part 1)**
>
> ### **(W1) Overcomplicated Design**
>
> Thank you for your thoughtful comments regarding the complexity of our model. We believe that the design elements are essential for effective driving reconstruction for several reasons:
>
> 1. **Necessity of Different Modules**: The 4D feedforward model requires the efficient integration of multiple perspectives and varying time intervals. To achieve this, we utilize temporal cross-attention to merge temporal information and implement the PD-Block for effective multi-view image fusion. Additionally, the GaussianAdapter facilitates the transfer of image features into a Gaussian representation.
> 2. **Importance of Regularization Terms**: At any given moment \( t \), the rendering supervision of the scene is limited by the sparse number of views. Moreover, the presence of multiple dynamic objects complicates monitoring at time \( t \). To address this, we decouple dynamic objects from static ones, allowing for better perspective utilization during the rendering process. Although this decoupling introduces numerous regularization terms, we regard them as necessary for achieving the desired outcomes.
> 3. **Significance of Pretrained Models**: Besides leveraging the SAM and DeepLab models, we acknowledge that more efficient methods exist for decoupling dynamic and static objects, which we discuss in detail in response to Q3. Importantly, DepthNet is not a pretrained network; it constitutes a part of our model that can be trained.
>
> Our findings in Table 4(a) demonstrate that these components are both valid and necessary. For downstream tasks, we utilize only the image encoder as the pretrained model, omitting the others to prevent any additional burden on training and inference. The model’s reconstruction inference speed is elaborated upon in (Q1).
>
> Furthermore, even widely recognized optimization-based reconstruction algorithms exhibit significant complexity in driving scenes [1, 2, 3, 4, 5]. They often integrate multiple annotations, regularizers, and neural network architectures.
>
> [1] Drivinggaussian: Composite gaussian splatting for surrounding dynamic autonomous driving scenes. *CVPR 2024*
>
> [2] Hugs: Holistic urban 3d scene understanding via gaussian splatting. *CVPR2024*
>
> [3] OmniRe: Omni Urban Scene Reconstruction. *arXiv preprint arXiv:2408.16760* (2024).
>
> [4] Street gaussians for modeling dynamic urban scenes. ECCV2025
>
> [5] Unisim: A neural closed-loop sensor simulator. CVPR2023.
>
> ### **(W2) The limation of realism**
>
> Currently, it appears that feedforward networks exhibit slightly lower rendering quality in driving scenarios compared to optimization methods. We have provided some reconstruction videos where you can observe our method at the following link: https://anonymize58426.github.io/Drive-Recon/. In the videos, you can see the lane lines being translated between 3 to 12 seconds, and the viewpoint being rotated between 17 to 26 seconds. This demonstrates that the scenes reconstructed by our algorithm maintain geometric consistency. I am confident that, in the near future, advancements in network architectures, training strategies, and generative techniques will enable feedforward networks to surpass optimization methods. This paper is  a preliminary study at the field of driving and give a code base, which will accelerate the development of the field.
>
> Furthermore, existing state-of-the-art generalizable Gaussian splatting algorithms often operate at relatively low resolutions and realism in indoor scenes [1, 2, 3]. Therefore, our paper is still at an advanced level in the field of generalizable Gaussian splatting.
>
> [1] Mvsplat: Efficient 3d gaussian splatting from sparse multi-view images. *ECCV* 2025.
>
> [2] Large Spatial Model: End-to-end Unposed Images to Semantic 3D. NeurIPS 2024
>
> [3] FreeSplat: Generalizable 3D Gaussian Splatting Towards Free-View Synthesis of Indoor Scenes Reconstruction. NeurIPS 2024

---

> ### Author Response · Authors · 2024-11-22
> **Author Response to Reviewer conn (Part 2)**
>
> ### **(Q1) Comparison to Optimization-Based Approaches**
>
> We appreciate your inquiry regarding efficiency. We require only 20 hours to complete 50,000 iterations on 24 A100 GPUs. According to the protocol outlined in Table 1, we have compared the time cost and PSNR of our method with that of traditional optimization methods.
>
> |  | PSNR | SSIM | LPIPS | Time Cost |
> | --- | --- | --- | --- | --- |
> | 3D-GS | 24.91 | 0.71 | 0.16 | 5.5h |
> | DrivingGaussian | 26.12 | 0.74 | 0.13 | 6.2h |
> | Ours | 23.70 | 0.68 | 0.17 | 1.21s |
>
> As illustrated in the table above, our algorithm is nearly comparable to traditional optimization methods, while inference takes only 1.21 seconds, highlighting our method's substantial speed advantage. The experiment was updated to part C of supplementary materials.
>
> ---
>
> ### **(Q2) The Need for 3D Boxes**
>
> I am grateful for your question about the use of 3D boxes. Our method can indeed be trained without requiring semantic segmentation or 3D boxes. As shown in Table 4(a), we can learn geometry through perspective consistency and point cloud depth information without the need for Dynamic and Static Rendering (DS-R).
>
> Additionally, we can use simple techniques for segmentation without utilizing 3D boxes. For example, we can compute optical flow from videos to identify points in dynamic regions, which can be used as prompts for the SAM to segment dynamic objects, enabling us to segment any type of dynamic object without 3D boxes.
>
> ---
>
> ### **(Q3) Downstream Applications**
>
> We train DriveRecon on the nuScenes training set without relying on 3D boxes or segmentation ($\lambda_{sr} =0$ and $\lambda_{seg}=0$). The pretrained model is then employed as an image encoder for UniAD. Then, UniAD is fine-tuned entirely using the original UniAD's training parameters. The results for UniAD in Table 6 are sourced from the original paper, and our pretrained model significantly enhances performance. I will provide clearer details about the experimental setup in the the paper.
>
> ---
>
> ### **(Q4) Miscellaneous**
>
> Thank you for your attention to our experimental setup. We utilized three frames of images as input for all experiments. In Table 4(b), "Training Num" refers to the mean number of scenes. Tables 1, 2, 3, and 4(a) use only 64 scenes (NOTA-DS64). Specifically, we trained with 64 scenes (NOTA-DS64) and tested with 54 new scenes (Diversity-54), achieving satisfactory results. This indicates that our algorithm demonstrates good generalization performance even when trained on a small dataset. We will reiterate these details in the appropriate sections of the paper.
>
>
> Thank you for your attention to our responses.  If you have any further questions or if anything remains unclear, please don’t hesitate to let us know.  We would be more than happy to discuss your concerns in greater detail.

---

> ### Author Response · Authors · 2024-11-25
> **Discussion request**
>
> Dear Reviewer conn,
>
> I would like to express my sincere gratitude to you for your constructive comments. As the ICLR discussion phase is almost over, I wanted to kindly ask if there are any remaining questions or clarifications needed regarding our responses. Please feel free to reach out at any time.
>
> We would be truly grateful if you could consider raising our rating, as your support is crucial for the potential acceptance of our work.
>
> Best wishes,
>
> Authors

---

> ### Author Response · Authors · 2024-11-25
> **Futher Disscussion**
>
> Dear Reviewer conn,
>
> Thank you for carefully reading our response and providing further comments. This is our further reply:
>
> ### **(P1) Complicated Design**
>
> We acknowledge that our approach is complex. But for autonomous driving, it's hard to avoid. For learnable driving tasks, such as perception task, they also have multiple modules including temporal fusion model,  multi-view fusion model, image encoder model, image decoder and detection head [1, 2]. These methods, which were considered too complex, have already been deployed to run on actual vehicles. So, this is not a shortcoming to reject our paper. Moreover, the two algorithms you mentioned are also complex [3,4].
>
> [1] Exploring object-centric temporal modeling for efficient multi-view 3d object detection. CVPR2023
>
> [2] Bevformer: Learning bird’s-eye-view representation from multi-camera images via
> spatiotemporal transformers. ECCV2023
>
> [3] G3r: Gradient guided generalizable reconstruction.ECCV 2025.
>
> [4] SCube: Instant Large-Scale Scene Reconstruction using VoxSplats." arXiv preprint arXiv:2410.20030 (2024).
>
> ### **(P2) Other generalizable approaches**
>
> Thank you for mentioning the two new related papers. The quality of the two papers produced is not significantly better than ours. If you zoom in on Fig.4 both of Paper 3 and Paper 4, you find that their rendering quality is not good enough. Paper 3 and 4 are difficult to compare because of its lack of open source and complex design. Paper 4 appeared after we submitted the paper. In addition, we discuss them further:
>
> Paper [3] also lacks realism. In the presentation of his paper, his images (Fig.4) are very small. If you zoom in these pictures are very blurry. And he reported reasoning speeds of 31s and 123s for a single scence. His model could not even predict the 3D Gaussian representation directly, and he needed multiple iterations of the network to predict the 3D Gaussian representation. Besides, his method is not open source and very difficult to reproduce with his complex algorithm.
>
> Method [4] appears on arxiv after we commit ICLR. The visualization in Figure 4 in his paper is not much better than ours. Because we are working with multiple views of multiple times, the resolution of the image we render is lower due to GPU memory. His approach was to take only three images as input and split the model into two stages. This allows him to render more high-resolution images. His method can learn geometric features entirely by relying on dense point cloud supervision, which uses a set of point-cloud dense methods. I believe that our approach has more potential with small improvements: (1) Splitting different time images onto different GPUs and then merging timing features across GPUS, which is already widely used in video generation. In this way we can render more high-resolution images. (2) Our approach further makes good use of pre-trained video models. However, his approach relies on Voxel-based models, which severely limits his upper boundary. (3) The visual encoder of our method can be used as a pre-training model, while his model cannot be used at all.
>
> I believe that in the near future, advances in network architecture, training strategies, and generation techniques will enable feedforward networks to transcend optimization methods. For example: (1) Use pre-trained video encoders and decoders to improve the rendering quality. (2) Assign different time images to different GPUs and then perform feature fusion across GPUs, which is a common operation in the field of video generation. (3) Use more driving datasets to train stronger models.
>
> In addition, the most advanced generalizable Gaussian spatter algorithms available generally operate with relatively low resolution and realism in indoor scenes [5,6,7]. Therefore, this paper is still at an advanced level in the field of generalized Gaussian sputtering.
>
> [5] Mvsplat: Efficient 3d gaussian splatting from sparse multi-view images. *ECCV* 2025.
>
> [6] Large Spatial Model: End-to-end Unposed Images to Semantic 3D. NeurIPS 2024
>
> [7] FreeSplat: Generalizable 3D Gaussian Splatting Towards Free-View Synthesis of Indoor Scenes Reconstruction. NeurIPS 2024
>
> ### **UniAD experimental**
>
> I would like to express my sincere apologies for the inaccuracy of my reply. I copied the original data of this table from Table 10 from VIDAR[8] ranther than UniAD. The experimental parameters were completely carried out in accordance with page https://github.com/OpenDriveLab/ViDAR. Specifically, we just replaced the image encoder and everything else was exactly the same.
>
> [8] Visual Point Cloud Forecasting enables Scalable Autonomous Driving. CVPR 2024
>
> Thank you for your attention to our responses. If you have any further questions or if anything remains unclear, please don’t hesitate to let us know. We would be more than happy to discuss your concerns in greater detail.

---

### Author Response · Authors · 2024-11-23
**General Response**

We fixed some minor typos and uploaded the paper. If you have any further questions or if anything remains unclear,  please don’t hesitate to let us know. We would be more than happy to discuss your concerns in greater detail.

We have made minor revisions to the paper, all of which are summarized as follows:

**Efficiency comparison:** Our approach is compared with both traditional optimization methods and the latest generalized feedforward networks in terms of latency and memory usage. Results and discussion are in section C of supplementary material.

**Detailed experimental process:** We have fixed some incorrect typos. We give more details of the pre-trained model and the image editing experiment.  We will proofread the paper thoroughly to enhance its writing, presentation, and layout.


Best wishes,

Authors

---

### Author Response · Authors · 2024-12-01
**Request for further feedback**

Dear Reviewers,

We sincerely apologize for troubling you once again, and we deeply appreciate the time and effort you have put into reviewing our submission. We responded further and look forward to discussing with you. As the ICLR discussion period has been extended, we still have approximately some days to continue the discussion. Please let us know if there are any additional points or concerns that you would like us to address before the discussion phase concludes. Your valuable feedback is highly appreciated and will help us further improve our work.

Best regards,

Authors

---

### Meta-Review · Area_Chair_MxhF · 2024-12-18

**Metareview:**

This paper proposes a driving scene reconstruction model, named DRIVINGRECON, which directly predicts 4D Gaussians from surround-view videos. While the idea is somewhat novel and offers a unique approach to scene reconstruction, the performance falls short compared to existing optimized methods. Additionally, the overall pipeline is excessively complex, which may hinder practical implementation and maintenance. Based on these strengths and weaknesses, the decision is not to recommend acceptance at this time.

**Additional Comments On Reviewer Discussion:**

This paper was reviewed by three experts in the field and finally received marginal scores of 5, 5, and 6.
Two major concerns of the reviewers are:
1.	the proposed method cannot achieve comparable performance relative to optimized methods,
2.	the pipeline is excessively complex.
The authors failed to address these two concerns during the discussion period.
I fully agree with these two concerns and, therefore, make the decision to reject the paper.

---

### Decision · Program_Chairs · 2025-01-22

Reject